# CooT 🐦: Learning to Coordinate In-Context with Coordination Transformers

## Abstract

Effective coordination among artificial agents in dynamic and uncertain environments remains a significant challenge in multi-agent systems. Existing approaches, such as self-play and population-based methods, either generalize poorly to unseen partners or require impractically extensive fine-tuning. To overcome these limitations, we propose Coordination Transformers (CooT), a novel in-context coordination framework that uses recent interaction histories to rapidly adapt to unseen partners. Unlike prior approaches that primarily aim to diversify training partners, CooT explicitly focuses on adapting to new partner behaviors by predicting actions aligned with observed interactions. Trained on trajectories collected from diverse pairs of agents with complementary preferences, CooT quickly learns effective coordination strategies without explicit supervision or parameter updates. Across diverse coordination tasks in Overcooked, CooT consistently outperforms baselines including population-based approaches, gradient-based fine-tuning, and a Meta-RL-inspired contextual adaptation method. Notably, fine-tuning proves unstable and ineffective, while Meta-RL struggles to achieve reliable coordination. By contrast, CooT achieves stable, rapid in-context adaptation and is consistently ranked the most effective collaborator in human evaluations.

## 1 Introduction

Coordination is fundamental to intelligent behavior, enabling individuals to achieve shared goals through joint effort. In dynamic and uncertain environments, such as team sports or traffic navigation, humans adjust their actions based on others' behaviors and intentions, facilitating effective collaboration. Replicating this adaptive coordination in artificial agents remains a core challenge in multi-agent systems (Cao et al., 2012; Zhang et al., 2021), especially in domains like robotics (Yan et al., 2013), gaming (Matignon et al., 2012), and human-AI interaction (Carroll et al., 2019), where an agent's success often depends on effectively responding to its partners.

Existing methods for developing agents capable of coordinating with unseen partners have pursued various strategies. Self-play (Tesauro, 1994; Yu et al., 2022a; Hu et al., 2021; Wang et al., 2023) trains agents by having them repeatedly interact with copies of themselves. Although this is effective in coordinating with known partners, it frequently leads to conventions that fail when interacting with unfamiliar collaborators. Population-based methods (Jaderberg et al., 2017) seek to address this by training agents within diverse populations using strategies such as partner randomization (Hu et al., 2020; Lucas & Allen, 2022), reward shaping (Yu et al., 2023), explicit modeling of partner behaviors (Lou et al., 2023), and enhancing behavioral diversity (Zhao et al., 2023; Lupu et al., 2021; Xue et al., 2024). However, while these approaches promote robustness, they often lack mechanisms for efficient online adaptation, which limits their applicability in open-ended or real-world scenarios. Attempting to fine-tune multi-agent reinforcement learning (MARL) policies on new partners also proves ineffective, since adaptation often demands thousands or even millions of interaction trajectories to learn effective coordination strategies (Nekoei et al., 2023), thus making few-shot adaptation infeasible.

Recent advances in in-context learning (ICL; Brown et al., 2020; Wei et al., 2022; Li et al., 2023) offer a promising alternative by enabling models to condition their behavior on contextual examples, without additional training or fine-tuning. Originally demonstrated in language modeling (Brown et al., 2020), ICL has recently been adapted to sequential decision-making domains, including offline

reinforcement learning (Chen et al., 2021; Lee et al., 2023) and distillation of reinforcement learning policies (Laskin et al., 2023; Kirsch et al., 2023). However, applying ICL to coordination introduces distinct challenges. Unlike task-focused settings with well-defined rewards, coordination requires aligning with diverse partner behaviors, often without explicit feedback or well-defined success metrics. Thus, the primary challenge shifts from generalizing across tasks to effectively interpreting and adapting to the behaviors of new partners purely based on observed interactions.

To address this challenge, we introduce Coordination Transformers (COOT), a framework designed specifically for in-context partner adaptation. Whereas prior ICL-inspired methods (Chen et al., 2021; Laskin et al., 2023) primarily target task generalization, COOT emphasizes generalization across diverse partner behaviors, a challenge less directly addressed in existing work. Specifically, COOT predicts actions that best align with the observed partner's behavior to maximize collaboration effectiveness. To achieve this, we train COOT on trajectories collected from interactions between pairs of agents whose behaviors reflect distinct underlying preferences. Such agents, which we term behavior-preferring agents, operate based on event-based hidden reward functions that guide their actions but are hidden from their partners. These event-based preferences yield diverse and potentially unpredictable coordination behaviors, even within identical environments. By observing interactions between these agents and their complementary counterparts, COOT learns context-driven strategies that enable effective coordination with previously unseen partners.

We evaluated COOT across a broad set of coordination methods, including population-based approaches (Yu et al., 2023; Zhao et al., 2023), gradient-based fine-tuning, and a meta-RL-inspired baseline (Rakelly et al., 2019) with contextual adaptation, using the Overcooked environment (Carroll et al., 2019), a popular multi-agent benchmark requiring precise and adaptive coordination. Results demonstrate that COOT consistently outperforms these baselines in diverse coordination tasks involving unseen agents. Notably, gradient-based fine-tuning proved unstable and delivered limited improvements, while the contextual adaptation Meta-RL baseline performed poorly, failing to achieve reliable coordination. By contrast, COOT achieves stable, rapid adaptation in context without parameter updates. Furthermore, human evaluations consistently rank COOT as the most preferred partner in interactive collaboration. Ablation studies provide additional insights, revealing that COOT improves coordination by observing more interactions with novel partners and that preserving temporal structure in interaction contexts is critical for generalization. Overall, these results demonstrate COOT 's effectiveness for adapting to diverse, unseen partners in cooperative settings.

## 2 RELATED WORK

**Learning to Coordinate.** A fundamental challenge in multi-agent reinforcement learning (MARL) is learning strategies that remain effective when paired with new partners. Self-play (SP) (Tesauro, 1994; Yu et al., 2022a; Wang et al., 2023) can produce strong agents, but independently trained SP policies often converge to *different conventions*, causing severe failures in cross-play (Carroll et al., 2019). This observation motivates research on how agents should coordinate with partners they have never trained with.

One line of work, known as zero-shot coordination (ZSC), formalizes this question under the assumption that all players are *independent instantiations of the same learning algorithm* (Hu et al., 2020; Treutlein et al., 2021; Hu et al., 2021). ZSC investigates which learning rules lead agents to adopt *compatible conventions* across separate training runs, especially in environments with symmetries where independent agents may otherwise diverge.

A broader setting is ad-hoc teamwork (AHT) (Stone et al., 2010), where an agent must coordinate with heterogeneous teammates whose training processes may differ. To improve robustness in this setting, population-based methods (Jaderberg et al., 2017; Long* et al., 2020; Hu et al., 2020; Strouse et al., 2021) expose agents to diverse partners, with further extensions based on entropy maximization (Zhao et al., 2023; Lupu et al., 2021), reward shaping (Lucas & Allen, 2022), structured population groups (Xue et al., 2024), hidden-utility biases (Yu et al., 2023). Recent work also explored training agents across large sets of procedurally generated environments (Jha et al., 2025), encouraging general cooperative behaviors that transfer to new partners and tasks. In parallel, the rise of large language models has led to LLM-assisted coordination methods (Zhang et al., 2024; Li et al., 2025), which leverage natural-language reasoning or high-level guidance to improve adaptivity.

Beyond cooperative teamwork, related work includes competitve or socially structured settings, see Appendix F for further details.

Despite these advances in ZSC and AHT, a key limitation remains: most approaches rely on large amounts of training interaction or computational cost to cover diverse partner behaviors. Few-shot coordination (Nekoei et al., 2023) instead seeks agents that can adapt to novel partners with minimal experience. Methods such as PACE (Ma et al., 2024) and PECAN (Lou et al., 2023) infer partner behavior from limited trajectories, while LIAM (Papoudakis et al., 2021) learns partner representations from local interaction histories to support rapid adaptation.

Building on this direction, our method COOT leverages the in-context learning capabilities of transformers to achieve effective few-shot coordination with diverse, previously unseen partners. Unlike approaches that first infer a latent partner identity (e.g., PACE, PECAN), COOT learns from full interaction histories to directly predict actions that best complement the observed partner's behavior, enabling efficient adaptation at test time without parameter updates.

**In-context Reinforcement Learning.** In-context learning (ICL) (Brown et al., 2020; Wei et al., 2022; Li et al., 2023) enables models to adapt at inference by conditioning on examples rather than gradient updates. Beyond its success in language and vision (Brown et al., 2020; Yu et al., 2022b), recent studies extend ICL to reinforcement learning (Chen et al., 2021; Jing et al., 2023; Laskin et al., 2023; Lee et al., 2023), showing applications in offline RL, online decision-making, opponent modeling, and meta-RL. Prior works suggest that transformers may implicitly implement optimization-like procedures (Von Oswald et al., 2023), with trajectories serving as contextual examples for learning algorithms or posterior sampling (Laskin et al., 2023; Lee et al., 2023; Jing et al., 2023). Unlike these task-focused approaches, COOT targets coordination, generalizing to unseen partners by interpreting prior interaction trajectories as context.

**Transfer Learning in Reinforcement Learning.** Transfer learning in reinforcement learning (RL) (Duan et al., 2016; Gupta et al., 2017; Finn et al., 2017; Tobin et al., 2017; Zhang et al., 2020; Xing et al., 2021) seeks to reuse knowledge from prior tasks to improve performance on new ones with minimal training, a key goal when task-specific experience is costly. Approaches include task-invariant representations (Gupta et al., 2017; Xing et al., 2021), meta-learning for rapid adaptation (Duan et al., 2016; Finn et al., 2017; Rakelly et al., 2019), and domain randomization (Tobin et al., 2017). Recent work highlights conceptual links to ICL (Laskin et al., 2023; Lee et al., 2023): both aim to generalize across conditions, but many transfer RL works rely on explicit parameter updates, while ICL adapts a fixed model by conditioning on interaction histories. This implicit form of transfer enables flexible, sample-efficient coordination. Building on this, COOT learns offline representations and adapts to novel online partners within only a few episodes.

## 3 PRELIMINARY

### 3.1 HIDDEN-UTILITY MARKOV GAME

We formulate the in-context coordination problem as a Hidden-Utility Markov Game (HU-MG), inspired by the framework introduced in Hidden-Utility Self-Play (Yu et al., 2023). The HU-MG builds on the two-agent decentralized Markov game (Bernstein et al., 2002), defined as the tuple $(\mathcal{S}, \mathcal{A}, \mathcal{T}, \mathcal{R}_t)$, where $\mathcal{S}$ denotes the state space, $\mathcal{A} = \mathcal{A}^i \times \mathcal{A}^w$ is the joint action space, $\mathcal{T} : \mathcal{S} \times \mathcal{A}$ represents the transition function, and $\mathcal{R}_t$ the shared global reward. Both agents, $\pi_i$ and $\pi_w$, aim to maximize $\mathcal{R}_t$.

A two-agent HU-MG is instead defined as $(\mathcal{S}, \mathcal{A}, \mathcal{T}, \mathcal{R}_t, \mathcal{R}_w)$, where $\mathcal{R}_w$ is a hidden reward space containing functions $r^w \sim \mathcal{R}_w$. The hidden reward $r^w$, observable only by $\pi_w$, is formulated as linear functions over event features, inducing distinctive behavioral preferences or conventions. Note that $\pi_w$ corresponds to the partner policy $\pi_i^{\mathrm{p}}$ introduced in Section 4, but we adopt the new notation to better distinguish partner indices in the dataset setting.

### 3.2 IN-CONTEXT LEARNING WITH DECISION-PRETRAINED TRANSFORMERS

In-context learning refers to a model's ability to generalize by conditioning predictions on contextual examples at inference time. A Decision-Pretrained Transformer (DPT) (Lee et al., 2023) trains on

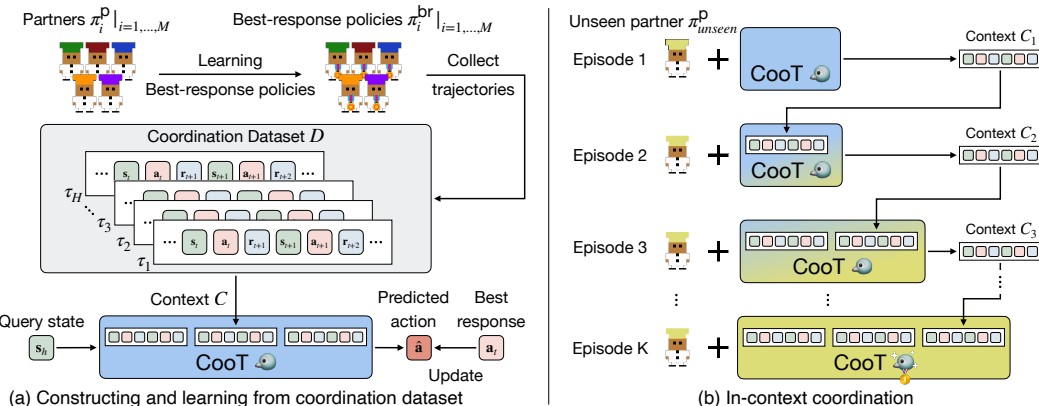

Figure 1: **COOT. (a) Training.** We generate a dataset $\mathcal{D}$ of trajectories between behavior-preferring agents and their best-response (BR) policies. For each training instance, COOT receives query states $\mathbf{s}_h$ and context $\mathbf{C}$ of past interactions, and learns to predict an action $\mathbf{a}$ mimicking the BR action $\hat{\mathbf{a}}$. **(b) Evaluation.** At test time, COOT coordinates with unseen partners by continually updating its context from recent episodes, which adapts to the partner online without gradient updates, enabling few-shot generalization through context update alone.

sequences of $(s, a, r)$ triplets, representing states, actions, and returns. At inference, it autoregressively predicts the optimal action for a given state, conditioned on contextual examples. This reframes reinforcement learning as sequence modeling, where task identity is inferred from context. However, standard DPT assumes access to extrinsic rewards and explicit task objectives, limiting applicability in coordination settings with diverse and implicit partner preferences.

## 4 METHOD

To build agents capable of efficient and adaptive coordination with unseen partners, we propose Coordination Transformer (COOT 🎯), a framework that adapts to novel partners by conditioning on past interactions. Our approach involves training a transformer on trajectories collected from diverse partners and their corresponding best responses, i.e., the behavior that yields the highest expected return given a specific partner's behavior, enabling effective generalization to previously unseen collaborators. We begin by formalizing coordination as a Hidden-Utility Markov Game (HU-MG) in Section 4.1, where agents behave according to hidden, agent-specific reward functions. We then describe the process of generating a diverse dataset of interaction histories in Section 4.2, explain how COOT is trained to predict partner-aligned actions in Section 4.3, and present our deployment protocol for evaluating generalization in Section 4.4. An overview of COOT is illustrated in Figure 1.

### 4.1 FROM TASK GENERALIZATION TO COORDINATION GENERALIZATION

Recent advances in in-context reinforcement learning enable sequence models to generalize across tasks by conditioning on prior experiences. These methods assume explicit task definitions and observable rewards, which allow models to infer goals or dynamics from context. However, they do not address the challenge of adapting to diverse partners with hidden preferences under the same task.

We reframe this challenge as a two-player Hidden-Utility Markov Game (HU-MG), where one of the agent's behaviors are driven by a hidden reward function that is unobservable to its partner. We refer to such agents as *behavior-preferring agents*, since their preferences are revealed only through behavior. These hidden utilities induce diverse coordination patterns, even when the environment remains fixed. In this setting, adapting to a partner requires inferring its underlying preferences through observed behavior.

Our approach treats partner adaptation as an in-context learning problem. Rather than relying on explicit task labels or extrinsic rewards, the model conditions on past interactions to infer hidden

utilities. In contrast to ICRL, where the goal is to recover the optimal policy for each task, in HU-MG, the objective becomes learning a *best-response policy* tailored to the inferred partner behavior.

## 4.2 DATASET GENERATION

To enable in-context adaptation to novel partners, we construct a dataset $D$ containing interaction histories between behavior-preferring agents and their corresponding best responses. Our data generation process is grounded in the HU-MG framework as described in Section 3.1, where coordination challenges arise from agent-specific hidden reward differences.

Following ZSC-Eval (Wang et al., 2024), we simulate hidden-utility preferences by first defining a set of discrete environmental events. We then construct different hidden reward functions using linear combinations of the environmental events to form a space of reward $\mathcal{R}_w$. For each hidden reward function $r_i^w \sim \mathcal{R}_w$, we jointly train a behavior-preferring partner policy $\pi_i^\mathrm{p}$, which is driven by $r_i^w$, and its best-response policy $\pi_i^\mathrm{br}$ using Proximal Policy Optimization (PPO) (Schulman et al., 2017).

To ensure diversity among training partners, we compute an event-based diversity score $d_i$ for each best-response policy $\pi_i^\mathrm{br}$, and then select the top-$N$ diverse pairs to form the training pool $\Pi_\mathrm{train}$. For each selected pair $(\pi_i^\mathrm{p}, \pi_i^\mathrm{br})$, we collect $T$ trajectories to form a fixed-length context $C$. From each context, we sample query states $s_h$ and construct tuples of the form $(s_h, C, \hat{a})$, where $\hat{a} = \pi_i^\mathrm{br}(s_h)$ is the best-response action. These tuples are added to the dataset $D$ for training COOT.

To construct the evaluation set, we extract behavioral features $\phi_i$ from trajectories generated by each $\pi_i^\mathrm{br}$ and compute the population diversity. To efficiently select a diverse evaluation set, we then apply Determinantal Point Process sampling (Kulesza et al., 2012) over $\mathbf{K}$, a similarity matrix of behavioral feature set $\{\phi_i\}$, resulting in a behaviorally diverse evaluation set $\Pi_\mathrm{eval} \subset \Pi_0$. Further evaluation details are provided in Appendix B.6.

## 4.3 TRAINING

Given the dataset of context–query–action tuples, we train the Coordination Transformer (COOT) to perform best-response prediction via in-context learning. At each training step, the model receives query states $s_h$ and a context window $C$, consisting of past interactions between a behavior-preferring agent and its best response. Conditioned on this context, the transformer predicts the best-response action distribution $\hat{p}_h(\cdot) = M_\theta(\cdot \mid s_h, C)$.

The model is trained to minimize the cross-entropy loss: $\mathcal{L} = -\log \hat{p}_h(\hat{a})$ where $\hat{a} = \pi_i^\mathrm{br}(s_h)$ is the ground-truth best-response action given the query states. This objective encourages the model to map partner behaviors to effective responses, supporting generalization through in-context adaptation to unseen collaborators. Further training details are provided in Appendix B.2

## 4.4 ONLINE DEPLOYMENT

At test time, COOT is deployed with unseen behavior-preferring partner policies $\pi_{unseen}^\mathrm{p} \sim \Pi_\mathrm{eval}$ over multiple episodes of length $Z$. A fixed-length context $C$, composed of $T$ previously collected trajectories $\tau = (s_t, a_t, r_t)_{t=1}^Z$, is initialized before deployment begins.

At each timestep $t$ in an episode of length $Z$, the agent observes the context $C$, current state $s_t$, and a short history of prior states (e.g., $\{s_{t-n}, \ldots, s_t\}$), which together form the query state $s_h$. Based on this input, COOT predicts an action distribution $\hat{p}_t(\cdot) = M_\theta(\cdot \mid s_h, C)$ and samples an action $a_t \sim \hat{p}_t$.

After executing action $a_t$ and observing reward $r_t$, the transition $(s_t, a_t, r_t)$ is recorded. Once the episode is complete, the resulting trajectory of $Z$ steps is appended to the context buffer. Repeating this process across $E$ episodes allows COOT to refine its coordination strategy through in-context adaptation alone. Without parameter updates, it progressively adapts to partner behaviors using recent interaction histories, demonstrating few-shot generalization in multi-agent coordination.

## 5 EXPERIMENTS

### 5.1 EVALUATION SETUP

**Environments.** We evaluate our method in the Overcooked environment (Lauffer et al., 2023), a cooperative multi-agent benchmark where two agents prepare and deliver soups under a fixed time limit. Agents can move, wait, or interact with ingredients (e.g., onions, tomatoes) and objects (e.g., pots, plates, dispensers), receiving a sparse reward of 20 for each successful delivery. To assess generality, we test across five layouts—*Asymm. Adv.*, *Bothway Coord.*, *Blocked Coord.*, *Coord. Ring*, and a multi-recipe variant of it. We evaluate these layouts' performance over 10 episodes per layout, except for the multi-recipe variant, where we use 15 episodes, as we follow the setup from ZSC-Eval (Wang et al., 2024). Each layout introduces distinct coordination challenges through resource distribution and spatial constraints. For example, the *Bothway Coord.* layout spatially separates work areas, preventing collisions while restricting plate access to a single agent. This asymmetry enforces a role division, with one agent providing plates and the other delivering soups, creating a collision-free coordination scenario. By contrast, the *Asymm. Adv.* layout distributes resources more evenly across zones, allowing agents to operate largely independently. This balanced structure minimizes the need for close coordination, making it a low-interaction scenario.

We compare COOT against various baselines.

- **Behavior cloning (BC)** learns to imitate expert demonstrations via supervised learning; it lacks mechanisms to adapt to different partners and is used as a non-aware partner baseline.

- **Maximum entropy population training (MEP; Zhao et al., 2023)** promotes behavioral diversity by regularizing policies with an entropy objective during population training, improving robustness to unfamiliar partners.

- **Hidden-utility self-play (HSP; Yu et al., 2023)** induces behavioral diversity by training agents with varied hidden reward functions, improving generalization to unseen partners. Both MEP and HSP are evaluated in dense-reward variants that use reward shaping (e.g., credits for ingredient pickups or placements) to stabilize training. While such shaping is unrealistic in human–AI collaboration, it yields stronger policies and thus provides an upper-bound comparison, whereas COOT is trained only with sparse rewards.

- **HSP-fine-tuning (HSP-ft)** fine-tunes HSP agents on novel partners to test whether gradient-based adaptation can improve coordination. Learning rates and last-layer updates are varied, following common fine-tuning practices.

- **HSP-meta** extends HSP with a trajectory encoder inspired by PEARL (Rakelly et al., 2019), an off-policy meta-RL method that learns to quickly adapt. The encoder, optimized with reconstruction loss, encodes recent episodes into a latent context appended to the observation, enabling adaptation to partner behaviors. Since HSP consistently outperforms MEP, this extension provides a stronger and more representative baseline.

**Evaluation pipeline.** Our evaluation follows ZSC-Eval (Wang et al., 2024), which measures ad-hoc teamwork capability with unseen partners. Specifically, evaluation partners are policies trained under diverse hidden reward functions, and a representative subset is selected using Best Response Diversity (the determinant of the similarity matrix of their best responses). Unlike the original protocol, which evaluates across multiple training checkpoints, we use only fully converged checkpoints to remove variability from partially trained behaviors and isolate final performance. Additional details are provided in Appendix B.6.

**Evaluation metrics.** We report Best Response Proximity (BR-prox) and average episode reward. BR-prox (Wang et al., 2024) quantifies how close an agent comes to the ideal best-response performance when paired with a given evaluation partner. More precisely, BR-prox is defined as the ratio between the agent's return when paired with a partner and the return that partner achieves with its own best-response teammate. A higher ratio indicates stronger coordination and generalization. To avoid degenerate cases, we exclude partners whose best-response return is zero. Average episode reward measures overall effectiveness across partners and complements BR-prox. We average both metrics over 50 rollouts per partner.

Table 1: **Benchmark results: CₒₒT outperforms baselines in coordination-heavy layouts.** We report the average episode reward (↑) and BR-prox (↑), both averaged across different layouts. For each method, we run three training seeds. Each seed is evaluated with 50 rollouts, and the resulting per-seed averages are used to compute the final mean ± std reported in the table. We bold results that lie within the best method's standard deviation range for each layout. CₒₒT maintains strong performance across all settings, with clear advantages in coordination-heavy layouts such as Multi-recipe and Counter Circ.

| Layout | Coord. Ring | | Coord. Ring Multi-recipe | | Counter Circ. | |
| --- | --- | --- | --- | --- | --- | --- |
| | Reward | BR-prox | Reward | BR-prox | Reward | BR-prox |
| BC | 26.24±1.80 | 0.31±0.02 | 8.97±0.49 | 0.10±0.01 | 10.79±5.33 | 0.11±0.06 |
| MEP | **40.30±3.45** | **0.47±0.04** | 16.64±1.16 | 0.19±0.02 | 1.89±0.41 | 0.02±0.00 |
| HSP | **41.10±10.03** | **0.49±0.10** | 29.35±3.77 | 0.33±0.04 | 21.37±2.17 | 0.23±0.03 |
| HSP-ft | **41.30±9.85** | **0.49±0.10** | 29.24±3.75 | 0.33±0.04 | 21.15±1.53 | 0.22±0.02 |
| HSP-meta | 29.84±3.92 | 0.35±0.04 | 30.21±1.37 | 0.34±0.02 | 3.28±0.21 | 0.03±0.00 |
| CₒₒT (Ours) | 38.30±3.71 | **0.47±0.06** | **45.96±3.99** | **0.50±0.04** | **28.28±2.32** | **0.30±0.03** |

| Layout | Asymm. Adv. | | Bothway Coord. | | **Overall** | |
| --- | --- | --- | --- | --- | --- | --- |
| | Reward | BR-prox | Reward | BR-prox | Reward | BR-prox |
| BC | 108.83±6.13 | 0.53±0.03 | 98.99±1.30 | 0.94±0.01 | 50.76±2.02 | 0.40±0.02 |
| MEP | 127.44±5.66 | 0.61±0.03 | 22.76±5.59 | 0.20±0.05 | 41.81±0.79 | 0.30±0.00 |
| HSP | **131.78±3.49** | **0.63±0.02** | 54.99±3.56 | 0.53±0.03 | 55.72±2.48 | 0.44±0.02 |
| HSP-ft | **131.79±3.22** | **0.63±0.02** | 55.81±2.71 | 0.54±0.02 | 55.86±2.29 | 0.44±0.02 |
| HSP-meta | 113.16±12.72 | 0.54±0.06 | 20.44±4.59 | 0.20±0.05 | 40.19±2.28 | 0.29±0.01 |
| CₒₒT (Ours) | **129.48±9.34** | **0.62±0.05** | **101.93±1.00** | **0.96±0.01** | **68.79±2.33** | **0.57±0.02** |

## 5.2 COORDINATION PERFORMANCE ACROSS BENCHMARKS

We next present results with unseen policy partners, comparing CₒₒT against strong baselines across diverse Overcooked layouts.

CₒₒT delivers consistently strong results across diverse Overcooked layouts, with its advantage becoming more significant as coordination demands increase. As Table 1 shows, CₒₒT is competitive in Coord. Ring and establish a clearer lead in Multi-recipe, where agents must balance concurrent goals. As layout complexity increases, unseen partners exhibit more diverse behaviors, making trajectory-level context increasingly valuable. Because CooT is pretrained to predict best-response actions conditioned on full interaction histories, it naturally develops an in-context coordination capability: at inference time, it adapts its behavior by conditioning on recent partner cues without learning an explicit partner model or performing gradient updates.

BC performs reasonably in layouts where agents operate in separated zones and collisions are avoided, such as *Asymm. Adv.* and *Bothway Coord.*. However, it struggles once coordination requires navigation or adaptation, as it lacks any mechanism to model or respond to partner behavior.

MEP and HSP perform well in simpler settings, such as *Asymm. Adv.*, a low-interaction scenario requiring little coordination, and the compact *Coord. Ring*, which combines navigation and ingredient handoffs within a constrained space. Yet their reliance on fixed policy pools limits flexibility: they underperform in *Bothway Coord.*, which requires reliable role coordination, and collapse in multi-recipe layouts that demand tracking multiple goals. Population-based methods rely on a fixed, precomputed set of partner policies. As the coordination structure becomes more complex, this finite pool becomes less representative of the behaviors encountered during evaluation, limiting their ad-hoc teamwork ability. HSP-ft further highlights the limits of gradient-based adaptation. Its performance is highly sensitive to learning rates and shows little improvement across episodes, sometimes degrading in more complex scenarios (Table 12, Appendix C.3), underscoring the advantage of context-based adaptation.

Surprisingly, HSP with trajectory conditioning (HSP-meta) underperforms vanilla HSP. Training the trajectory context encoder jointly with the policy introduces challenges: early in learning, noisy or uninformative latents can mislead the policy, and the reconstruction loss may not align with coordination objectives. Subtle partner differences may also be overlooked when reconstructing raw trajectories, limiting the usefulness of the latent context. These results suggest that although trajectory-level conditioning is appealing, effective coordination requires more principled context representations, which are non-trivial to design.

Overall, CooT achieves robust coordination with policy-based evaluation partners. But do these gains transfer to human collaborators, whose behaviors are more varied and less predictable?

## 5.3 Human-Agent Collaboration Study

Unlike trained agents, humans display diverse styles and adapt unpredictably. To assess whether CooT 's advantages extend beyond controlled benchmarks, we conducted a user study in the *Coord. Ring* layout, which integrates navigation and ingredient coordination within a compact space to elicit diverse collaboration scenarios. A total of 36 participants each played eight rounds, blindly paired through random assignment with one of four representative agents: CooT, HSP, MEP, or BC. We excluded HSP-ft and HSP-meta since preliminary experiments showed they were not competitive and provided little additional insight; restricting the study to four agents also reduced participant fatigue. Each successful delivery contributed 20 points to the score. To ensure consistency, we fixed the time interval between rounds and excluded the top and bottom 10% of participant scores to reduce variability from fatigue or outliers. All participants provided informed consent, and the study was conducted in accordance with standard ethical guidelines. Further details on the human evaluation platform and experimental setup are provided in Appendix D.

Table 2: **Human evaluation returns.** Comparison of mean episode reward (after outlier removal) across methods. CooT achieves the highest return and consistently outperforms baselines, indicating stronger coordination with human partners.

| Method | Mean | Std | 25% | 50% | 75% |
|---|---|---|---|---|---|
| BC | 51.0 | 3.8 | 47.5 | 52.5 | 55.0 |
| MEP | 53.0 | 5.1 | 49.4 | 53.8 | 57.5 |
| HSP | 40.5 | 5.2 | 36.9 | 40.0 | 45.0 |
| CooT (Ours) | **63.5** | 9.5 | **56.9** | **62.5** | **68.1** |

Table 3: **Human evaluation ratings.** Human evaluations (1–6 scale, higher is better) for four different agents based on two abilities: Collaborative and Adaptive. Each cell contains the score given by a user. The overall score is the sum of the two individual scores.

| Method | Collab. | Adapt. | Overall |
|---|---|---|---|
| BC | 2.0 | 1.8 | 3.4 |
| MEP | 3.4 | 2.9 | 6.3 |
| HSP | 2.7 | 2.2 | 4.9 |
| CooT (Ours) | **4.0** | **3.1** | **7.1** |

**Quantitative results.** Table 2 reports average performance across participants after outlier removal. CooT achieved the highest mean score, outperforming all baselines by a large margin. Table 3 summarizes 1–6 ratings of collaboration and adaptivity, where CooT again achieved the best scores on both dimensions. Figure 2 further shows that CooT was most frequently selected as the preferred partner (18 of 36 participants). Together, these results demonstrate that CooT not only delivers the strongest objective outcomes but is also consistently favored in subjective human evaluations.

**Qualitative feedback.** Post-round questionnaires presented in Table 13 in Appendix D.3 highlight both strengths and limitations of CooT. Participants often noted improvement over time: "Initially it doesn't do well, but gradually the performance improves," and "smoother collaboration over time." Others pointed to supportive behaviors, such as "It collaborates with me well, doing work on its own and helping me with onion preparation." At the same time, some observed occasional blocking or rigidity, e.g., "It blocked me in the beginning, but then it adapted." Overall, CooT was viewed as a capable collaborator, though not always seamless.

**Discussion.** The human study shows that CooT performs strongly not only with policy-based partners but also with real humans, whose behaviors are more varied and less predictable. Notably, in the *Coord. Ring* layout where CooT was similar to agent partners, it achieved a clear margin when paired with human participants. These findings suggest that CooT is particularly effective at handling the variability and adaptation demands that arise in real interactions. To further investigate the source of this advantage, the next section analyzes two key properties of CooT: its ability to adapt rapidly to out-of-distribution behaviors and its robustness to non-stationary partners.

## 5.4 Analysis of adaptation dynamics

The human study suggests that CooT can handle the variability of real interactions, but the mechanisms behind this advantage remain unclear. To probe further, we analyze two key properties of CooT in controlled policy-based settings: its ability to adapt rapidly to new partners and its robustness to sudden changes in partner behavior.

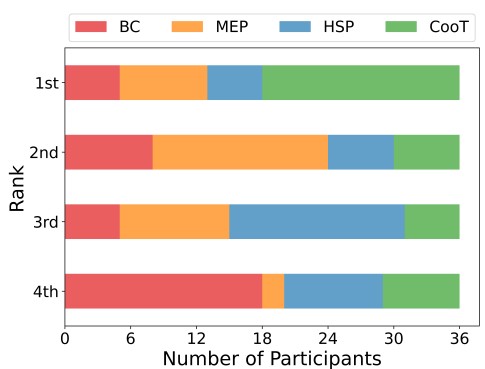 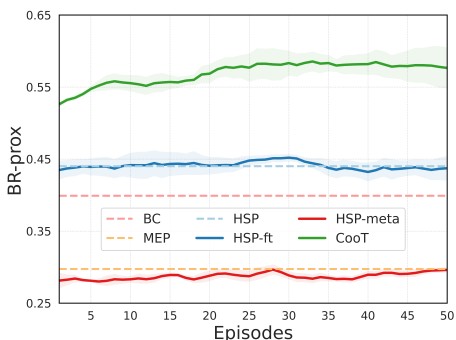

Figure 2: **Human study: agent ranking distribution.** Number of participants who chose each agent at different rankings. CooT received the highest number of first-place rankings, indicating it is the most preferred collaborator.

Figure 3: **In-context performance improvement of CooT over episodes.** As more partner trajectories are observed, CooT steadily improves its coordination strategy, highlighting the advantage of context-based adaptation.

Table 4: **Adaptation speed for non-stationary strategies.** Episodes needed for CooT to attain its average performance with the new partner.

| $P_A \to P_B$ | $P_{15} \to P_{17}$ | $P_{17} \to P_{15}$ | $P_{15} \to P_{31}$ | $P_{31} \to P_{15}$ | $P_{17} \to P_{31}$ | $P_{31} \to P_{17}$ |
|---|---|---|---|---|---|---|
| Episodes to adapt | 6 | 2 | 6 | 5 | 2 | 1 |

### 5.4.1 RAPID ADAPTATION THROUGH CONTEXT

We first measure how CooT adapts over time when paired with previously unseen partners. Figure 3 reports averages across seven layouts, each evaluated with 10–15 partners over 50 episodes.

CooT improves markedly within the first 5–8 episodes and continues to refine more gradually up to episode 50, showing that only a handful of trajectories are sufficient to align with a new partner. In contrast, the baseline methods show minimal improvement: HSP-ft exhibits unstable performance with occasional degradation, while HSP-meta demonstrates a rather slow adaptation.

These results highlight CooT 's ability for rapid context-based adaptation — a crucial property for human–AI collaboration, where effective coordination must be established quickly without prior exposure. Note that CooT performs adaptation at the episode level: the context buffer is updated only after each full trajectory, so the model maintains a consistent behavior within an episode.

### 5.4.2 ROBUSTNESS TO NON-STATIONARY PARTNERS

Human partners often change strategies mid-task. To test robustness to such non-stationarity, we conduct a controlled partner-swap experiment: CooT first plays several episodes with one HSP biased partner ($P_A$), accumulates context, and then is suddenly paired with a different biased partner ($P_B$) that follows a distinct preference. We then measure how many episodes it takes for CooT to recover the level of performance it typically achieves with the new partner when trained from scratch. The procedure is repeated in both directions ($P_A \to P_B$ and $P_B \to P_A$).

Results in Table 4 show that CooT adapts within at most six episodes, with an average of 3.67 episodes across partner switches. This demonstrates that CooT can rapidly recalibrate its coordination strategy when faced with abrupt partner changes, relying only on its context buffer without fine-tuning or retraining. Such robustness to non-stationary strategies is essential for real-world human–AI collaboration.

**Discussion.** Together, these results show that CooT can both adapt quickly to new partners and remain stable under abrupt partner changes—two properties that are essential for real-world coordination. As a final check, we perform ablations to verify which design factors contribute most to these capabilities.

While CooT does not update context mid-episode, the partner-swap results show that episode-level updates are sufficient to handle abrupt changes in partner strategy.

### 5.5 ABLATION STUDIES

To better understand which design choices are most critical for COOT, we perform small-scale ablations focusing on two factors: context length and trajectory shuffling.

**Context length.** Table 10 in Appendix C.1 shows that performance improves when conditioning on longer contexts, up to five episodes. Beyond this point, gains diminish while computation increases, so we adopt five episodes as the default.

**Trajectory shuffling.** Table 11 in Appendix C.2 shows that during training, step-wise shuffling disrupts temporal structure and harms performance, while chunk-wise shuffling has little effect. This confirms that preserving long-range dependencies in interaction histories is important.

## 6 CONCLUSION

We introduced Coordination Transformer (COOT 🌐), an in-context coordination framework that enables agents to adapt to unseen partners by conditioning on recent interactions. Unlike prior approaches emphasizing task generalization or large training populations, COOT targets partner generalization, predicting best-response actions without fine-tuning. Experiments on Overcooked show that COOT outperforms strong baselines with both evaluation agents and humans, while analyses highlight its rapid adaptation and robustness to unseen partners. Our main contributions are the formulation of partner-centric in-context coordination and extensive empirical validation in a challenging coordination domain. However, our work is limited by evaluation in a relatively structured environment and the lack of explicit modeling for mutual adaptation. Future work can address these limitations by extending COOT to more complex, co-adaptive, and real-world multi-agent settings.

### REPRODUCIBILITY STATEMENT

To ensure reproducibility, the Supplementary Material includes the training and evaluation scripts, together with configuration files and detailed instructions for reproducing the reported results.

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

APPENDIX

# Table of Contents

## A    EXPERIMENT ENVIRONMENT

### A.1    OVERCOOKED

We evaluted in the Overcooked environment (Carroll et al., 2019) inplemented in ZSC-Eval (Wang et al., 2024)[1]. We used five layouts, Bothway Coordination (*Bothway Coord.*), Coordination Ring (*Coord. Ring*), Counter Circuit (*Counter Circ.*), Asymmetric Advantages (*Asymm. Adv.*), and Coordination Ring with Multi-Recipe (*Coord. Ring multi-recipe*). The multi-recipe layouts have onion (O) and tomatoes (T) as ingredients, which expands the range of recipes from just onion soup (3O) to five types of soups, including mix soup (1O1T), less onion soup (2O), tomato-onion soup (2T1O), onion-tomato soup (2O1T), and onion soup (3O). The following are the details and main challenges for each layout.

**Bothway Coordination.** In this variant, both players can access onions and pots, which broadens the range of feasible strategies and introduces new opportunities for cooperation. This layout helps reduce idle time seen in the Forced Coordination setting and encourages more diverse policies. Still, since plates and the serving station remain confined to one side, effective teamwork is essential to fulfill orders.

---

[1] `https://github.com/sjtu-marl/ZSC-Eval`, with MIT License.

**Coordination Ring.** The Coordination Ring features a compact, circular layout that facilitates close agent interaction. Ingredients, plates, and the serving station are grouped in the bottom-left area, while cooking pots are placed in the top-right. This spatial arrangement drives continuous movement and requires players to coordinate effectively as they manage shared resources and navigate the kitchen.

**Counter Circuit.** Similar in shape to Coordination Ring, the Counter Circuit introduces a central, elongated table, creating narrow pathways that often cause congestion. This layout demands careful movement planning, as agents must avoid blocking one another. A common cooperative tactic involves staging onions in the center to streamline ingredient transfer.

**Asymmetric Advantages.** This layout divides the kitchen into two largely self-contained workspaces while maintaining interdependence through asymmetrically shared resources. Each player has unique access to ingredients and serving stations, while two centrally located pots are jointly accessible. Notably, one player benefits from closer proximity to the serving station, encouraging the development of collaboration strategies to balance workload and improve efficiency.

**Coordination Ring with Multi-Recipe.** This extended version of Coordination Ring includes tomatoes positioned near the serving area in the bottom-left corner, increasing task complexity. The added ingredient and recipe variety heighten the need for coordinated planning and amplify the importance of cooperation in fulfilling diverse orders.

## A.2 GOOGLE RESEARCH FOOTBALL

**3 vs 1 with Keeper.** In the scenario, three controlled left-team players start near the three edges (left, top, and bottom) of the right-team goal area. The player ($P_0$) standing near the left edge starts with the ball. The two other controlled players ($P_1$ and $P_2$) begin in open passing positions. And a right-team defender, attempting to block their attack, stands between $P_0$ and the goal, along with a right-team goalkeeper. This setup naturally requires multi-agent teamwork among three attackers and introduces longer-horizon coordination compared to Overcooked.

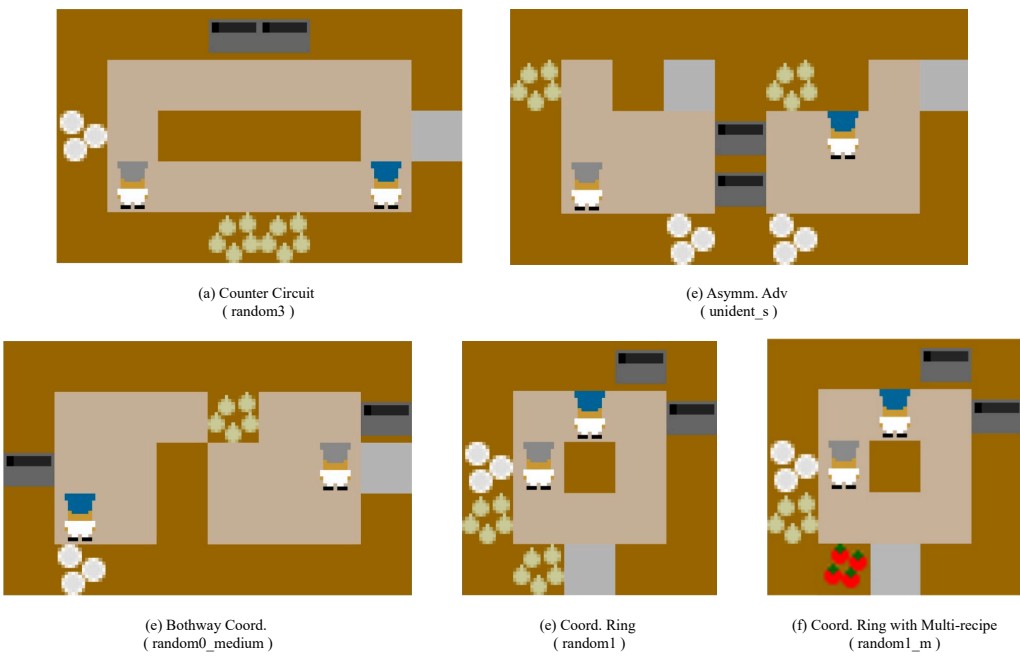

(a) Counter Circuit
( random3 )

(e) Asymm. Adv
( unident_s )

(e) Bothway Coord.
( random0_medium )

(e) Coord. Ring
( random1 )

(f) Coord. Ring with Multi-recipe
( random1_m )

Figure 4: Used layouts in Overcooked.

## B  IMPLEMENTATION AND DATASET DETAILS

### B.1  SOURCE AND LICENSING OF POLICY POOLS AND DATASETS

**HSP and MEP training datasets.**   In our setting, we adopt behavior-preferring rewards, which refer to event-based biased reward functions. These rewards assign credit to specific events, such as picking up onions or put an ingredient into pots. We assume that the behavioral requirements of deployment-time partners can be effectively modeled using this set of reward functions. Details about the reward specifications can be found in Table 5. We defined different sets of event-based rewards according to the characteristics of each layout. For example, the biased agent in Bothway Coordination cannot deliver soup, so we shifted the biased reward from delivering soup to picking up onions or dishes in order to obtain sufficient biased agent policies. In contrast, the biased agent in the multi-recipe Coordination Ring can access not only onions but also tomatoes, so some of the events we selected are related to tomatoes.

For two-stage algorithms such as HSP (Yu et al., 2023) and MEP (Zhao et al., 2023), while we do not train these policies ourselves, we construct the candidate policy pool by loading pretrained checkpoints from the ZSC-eval repository, which contains a diverse collection of agents trained under different bias reward settings and algorithmic configurations. These policies exhibit varied behaviors and serve as potential partners for training adaptive agents in the second stage. Following the settings in ZSC-eval (Wang et al., 2024), we selected 36 candidate policies as the basis for partner selection.

Table 5: **Events and biased reward under different layouts.** Each entry lists all possible biased reward values that can occur for the corresponding event under the given layout type.

| **Events** | **Bothway Coord.** | **Multi-recipe** | **Others** |
|---|---|---|---|
| Pickup onions from dispensers | -20,0,10 | 0 | -20,0,10 |
| Pickup dishes from dish dispensers | 0,10 | 0 | -20,0,10 |
| Pickup onion or dish from counters | -20,0 | 0 | 0 |
| Pickup soup from counters | -20,0 | 0 | 0 |
| Put onions into pots | 0 | -20,0 | 0 |
| Put tomatos into pots | | -20,0 | |
| Deliver soup with two ingredients | | -5,0,20 | |
| Deliver soup with three ingredients | | -15,0,10 | |
| Deliver soup | 0 | 0 | -20,0 |
| Stay | -0.1,0,0.1 | -0.1,0,0.1 | -0.1,0,0.1 |
| Order Reward | x0.1,x1 | x1 | x0.1,x1 |

**COOT and BC training datasets.**   We construct our expert dataset mainly following the same procedure as in HSP. To build a robust and diverse partner pool $\Pi_{\text{train}}$ for training adaptive policies, we include 36 agents sourced from both MEP and HSP. Specifically, we collect 15 MEP agents with final skill levels, and 21 HSP agents greedily selected based on an event-based diversity score $d_i$, which measures the expected frequency of key events.

For each selected pair $(\pi_i^{\text{p}}, \pi_i^{\text{br}}) \in \Pi_{\text{train}}$, we collect $J$ joint trajectories $\tau$, where $J = 200$ for MEP agents and $J = 250$ for HSP agents (including 220 from final-skill-level and 30 from mid-skill-level checkpoints). While COOT utilizes the current state $s_t$ and the context $C$ to predict $\hat{a}$, the BC baseline is trained using the same dataset by conditioning only on the state $s_t$. Further details of implementation and training will be discussed in the upcoming paragraphs.

### B.2  COOT TRAINING DETAILS

**Generation of training dataset.**   To train our Coordination Transformer (COOT), we begin with a set of pretrained policy pairs consisting of biased partner agents and their corresponding best-response policies. For each policy pair, we first select one pair and sample $T$ episodes of rollouts, which we use to construct a context $C$. This process is repeated $K$ times per policy pair to generate $K$ distinct contexts. For each context $C$, we then sample $L$ different query states $s_h$ from the rollout trajectories. Each query state, combined with its associated context $C$ and the corresponding optimal

action $\hat{a}$, forms a single training data point of the form $(s_h, C, \hat{a})$. Given a total of $M$ policy pairs, this procedure results in a dataset containing $M \times K \times L$ training examples. More detailed information can be found in Table 6.

**Online deployment.** During online deployment, COOT operates with a dynamically updated context $C$ to enable in-context adaptation. At the start of the first episode, $C$ is initialized as an empty context. Since no prior trajectory is available, CooT selects actions based solely on the current state and the empty context. After completing the first episode, the entire trajectory is stored and used to construct a new context, which is then appended to $C$. As a result, $C$ contains one populated trajectory and $T-1$ empty slots (assuming a fixed context length of $T$). In subsequent episodes, CooT utilizes the current buffer to condition its action predictions, allowing it to adapt based on accumulated experience. The context is managed using a first-in, first-out (FIFO) policy, ensuring that the most recent $T$ trajectories are always retained. This procedure is repeated iteratively throughout the evaluation phase to simulate continual adaptation in a coordination setting. The main method details are provided in Algorithm 1 and Algorithm 2. Algorithm 1 outlines the training process of COOT, while Algorithm 2 outlines the evaluation process, including how the evaluation partners are selected.

### B.3 BASELINE AND TRAINING IMPLEMENTATION

Our baseline codebase is primarily based on two open-source frameworks: Imitation[2], an imitation learning library built on Stable Baselines3[3], and ZSC-eval (Wang et al., 2024), a benchmark codebase for Zero-Shot Coordination (ZSC). We use the original implementation of Behavioral Cloning (BC) from Imitation, while the implementations and hyperparameter settings of HSP and MEP are directly inherited from ZSC-eval and HSP (Yu et al., 2023).

All models are evaluated in sparse-reward settings under ZSC environments to better reflect real deployment. To provide a more comprehensive comparison, we additionally report the performance of HSP and MEP trained with dense rewards, which offer frequent intermediate feedback (e.g., picking up onions, placing them in pots) to facilitate more stable and efficient training.

Table 6: **Dataset configuration for training the Coordination Transformer (CooT).** It specifies the number of partner policy pairs, contexts, rollouts, and resulting total dataset size.

| Dataset | Parameter | Value |
|---|---|---|
| | Number of partner policy pairs ($M$) | 36 |
| | Number of contexts per pair ($K$) | 125 |
| CooT Dataset | Number of rollouts per context ($T$) | 5 |
| | Number of data per context ($L$) | 70 / 200 (GRF) |
| | Total dataset size ($M \times K \times L$) | 315,000 |

---

[2]`https://github.com/HumanCompatibleAI/imitation`, with MIT license
[3]`https://github.com/DLR-RM/stable-baselines3`, with MIT license

Table 7: **Training hyperparameters for all methods.** We list model settings for CᴏᴏT, PPO parameters for HSP/MEP, and additional encoder parameters for HSP-meta.

| Method | Hyperparameter | Value |
|---|---|---|
| BC | Batch size | 256 |
| | Learning rate | 0.001 |
| | Optimizer | Adam |
| | Scheduler | CosineAnnealingLR |
| | Max Epochs | 50 |
| | Early Stopping Patience | 5 |
| | Validation Split | 0.1 |
| | Scheduler $\eta_{min}$ | $\frac{lr}{100}$ = 3e-6 |
| CᴏᴏT | Batch size | 120 |
| | Learning rate | 5e-5 |
| | Optimizer | Adam |
| | Scheduler | LambdaLR |
| | Weight decay | 1e-3 |
| | Dropout | 0.3 |
| | Gradient clip norm | 0.25 |
| | Model | GPT-2 |
| | Hidden layers | 4 |
| | Attention heads | 2 |
| | Embedding size | 128 |
| | Max Epochs | 70 |
| | Early Stopping Patience | 25 |
| Stage2 of HSP and MEP | Entropy coefficient | 0.01 |
| | Gradient clip norm | 10.0 |
| | GAE lambda | 0.95 |
| | Discount factor ($\gamma$) | 0.99 |
| | Value loss | Huber loss |
| | Huber delta | 10.0 |
| | Optimizer | Adam |
| | Optimizer epsilon | 1e-5 |
| | Learning rate | 5e-4 |
| | Parallel environment threads | 100 |
| | Environment steps | 10M |
| | Episode length | 200 |
| | Reward shaping horizon | 10M |
| | Policy pool size | 36 |
| HSP-meta | Parallel environment threads | 50 |
| | Encoder loss | Reconstruction loss |
| | Encoder layers | 3 |
| | Encoder lr | 1e-5 |

Table 8: **Computational resources used.** Specifications of the four workstations used for training and evaluation, including CPU, GPU, and memory.

| Workstation | CPU | GPU | RAM |
|---|---|---|---|
| Workstation 1 | Intel Xeon W-2255 | NVIDIA GeForce RTX 3080 | 125 GiB |
| Workstation 2 | Intel Xeon W-2255 | NVIDIA GeForce RTX 3090 ×2 | 125 GiB |
| Workstation 3 | Intel Xeon w7-2475X | NVIDIA GeForce RTX 4090 | 125 GiB |
| Workstation 4 | Intel Xeon w7-2475X | NVIDIA GeForce RTX 4090 ×2 | 125 GiB |

## B.4 Hardware Specifications and Training Time

We performed the experiments using the workstations listed in Table 8. Our method takes approximately 19 hours to train for each layout. Population-based baselines such as MEP and HSP require per-partner adaptation via online cross-training, which takes approximately 5 hours per partner with 100 parallel environment threads. Reproducing all results, including the baselines, requires approximately 300 GPU hours when run sequentially.

Table 9: **Average per-step inference time across model architectures.** Reported times reflect pure model forward pass per action, excluding environment interaction.

|  | CooT | MEP | HSP | BC |
|---|---|---|---|---|
| Inference Time (ms) | 2.41 | 1.73 | 1.77 | 0.54 |

## B.5 Inference Time Comparison

To complement the training cost reported above, we provide a comparison of pure model inference time, excluding environment interaction, across the architectures used in our experiments. Table 9 reports the average per-step inference time. While transformer-based ICL introduces additional overhead compared to lightweight baselines, the runtime remains practical in our multi-agent setting, as CooT requires only 2.41 ms per step, which is acceptable for our simulation-based evaluations.

## B.6 Evaluation Pipeline Details

To evaluate ad-hoc teamwork with unseen partners, we follow the ZSC-Eval framework (Wang et al., 2024), which constructs a diverse partner population and measures how well an ego agent coordinates with them. Below, we provide the implementation details omitted from the main text.

**Behavior feature extraction.** For each candidate partner $\pi_{\mathbf{w}}$, we compute a high-level behavior feature vector from event occurrences. We define

$$\phi(s_t, a_t) \in \mathbb{R}^m$$

as an *event-based feature vector*, where each component $\phi_j(s_t, a_t)$ indicates whether the $j$-th predefined event occurred at time step $t$ (e.g., pick-up, drop-off, interact-with-object, move-to-location). The overall behavior feature of the approximate best response is

$$\theta_{\mathbf{w}} = \mathbb{E}\left[\sum_{t=1}^{T} \phi(s_t, a_t)\right],$$

computed over full episodes with partner $\pi_{\mathbf{w}}$ and its best response.

**Similarity computation.** Similarity between two partners is defined as the dot product of their behavior features:

$$K_{ij} = \theta_i^\top \theta_j.$$

**Best Response Diversity.** For a subset of partners $\mathcal{S}$, we compute the Best Response Diversity as

$$\mathrm{BR\text{-}Div}(\mathcal{S}) = \det(K_{\mathcal{S}}),$$

where $K_{\mathcal{S}}$ is the similarity submatrix restricted to $\mathcal{S}$. Larger determinants correspond to subsets whose best responses exhibit more diverse behaviors.

**Partner selection via Determinantal Point Process.** The partner subsets are sampled using a Determinantal Point Process (DPP) with kernel $K$. Multiple subsets are drawn, and we select the one with the highest BR-Div as the evaluation partner set. Earlier training checkpoints of the selected partners are also included to capture a wider range of skill levels.

Table 10: **Effect of context length on coordination performance in *Coord. Ring*.** Performance improves steadily as context length increases from 1 to 5 episodes, but gains diminish when extended to 7 episodes.

| Context length | Reward (↑) |
|---|---|
| 1 episode | 33.87 |
| 3 episodes | 34.36 |
| 5 episodes | **41.11** |
| 7 episodes | 36.04 |

Table 11: **Average episode reward and BR-prox under different shuffle strategies on *Coord. Ring*.** Preserving temporal structure results in better coordination and improved adaptation performance.

| Shuffle strategy | Reward (↑) | BR-prox (↑) |
|---|---|---|
| No Shuffling | 37.47±5.67 | 0.44±0.094 |
| Step-wise | 29.41±5.23 | 0.36±0.084 |
| Chunk-wise | **38.30±3.71** | **0.47±0.056** |

## C  COMPLEMENTARY EXPERIMENTS

### C.1  CONTEXT LENGTH

Performance improves steadily as context length increases up to 5 episodes, reaching the best mean reward of 41.1. Beyond this point, gains diminish and even drop slightly at 7 episodes, indicating that overly long contexts may dilute relevant information while adding computational overhead. We therefore adopt 5 episodes as the default setting, balancing effectiveness and efficiency.

### C.2  CONTEXT SHUFFLING

While COOT demonstrates robust performance in unseen partner coordination, we investigate whether its generalization ability can be further enhanced through improved training-time augmentation strategies. Specifically, we explore whether trajectory augmentation can help the model acquire more robust coordination behaviors. Motivated by Decision Pretrained Transformers (DPT) (Lee et al., 2023), where step-wise trajectory shuffling enhanced generalization in partially observable navigation tasks, we examine whether similar augmentation methods benefit our multi-agent coordination setting. To this end, we evaluate two trajectory-shuffling strategies, step-wise and chunk-wise, in the *Coord. Ring* layout over 50 rollouts each. Although step-wise shuffling has shown benefits in simple navigation environments such as Dark-Room (Zintgraf et al., 2021), where agents operate under short horizons and limited observability, we observe a performance decline when applying this strategy in Overcooked. Unlike Dark-Room's short-horizon navigation tasks, Overcooked requires tight coordination over extended horizons. In such settings, preserving temporal structure remains critical, as effective coordination in Overcooked requires agents to act in a tightly timed relation to one another to avoid unnecessary delays or idle time.

As shown in Table 11, chunk-wise permutation, which shuffles multi-step segments while preserving key temporal dependencies, shows a slight improvement over step-wise shuffling and no augmentation. Interestingly, step-wise shuffling disrupts essential temporal continuity despite increasing data diversity and performs worse than unaugmented data. These findings emphasize the importance of preserving long-range temporal structure in trajectory augmentation, especially for multi-agent tasks that rely on temporally extended interactions.

### C.3  FINE-TUNING

Table 12 illustrate the instability of gradient-based adaptation in coordination settings. While large learning rates (e.g., $10^{-4}$) consistently destabilized training, leading to sharp performance drops, smaller learning rates ($10^{-7}$ and $10^{-8}$) yielded slight improvement. Partial fine-tuning (p-ft) often underperformed full updates, suggesting that restricting adaptation to the actor head fails to capture the dynamics needed for partner alignment. Overall, the limited and unstable gains from fine-tuning highlight the contrast with COOT.

### C.4  ADAPTATION

Figure 5 further highlights the performance gap between COOT and the HSP-Meta baseline. Across all five layouts, COOT generally improves with additional episodes, while HSP-Meta remains largely

Table 12: **Comparison of online fine-tuning baselines (MAPPO with different learning rates) against COOT.** Each model was evaluated with 50-episode interactions across 9 partner seeds. "p-ft" denotes updating only the actor head. Fine-tuning remains unstable: large learning rates cause collapse, while small rates yield no consistent gains.

| Partner | no ft | 1e-8 | 1e-7 | 1e-6 | 1e-5 | 1e-8 p-ft | 1e-7 p-ft | 1e-6 p-ft |
|---|---|---|---|---|---|---|---|---|
| 2 | **8.0** | 7.6 | 6.8 | 6.4 | 1.6 | **8.0** | **8.0** | 6.8 |
| 12 | 6.8 | 6.8 | 7.2 | **8.0** | 5.2 | 6.8 | 6.8 | 6.4 |
| 15 | 39.2 | 38.4 | 39.6 | **41.6** | 32.0 | 39.2 | 39.2 | 38.4 |
| 16 | 36.8 | **38.4** | **38.4** | 37.2 | 32.4 | 36.8 | 36.8 | 36.0 |
| 17 | 28.4 | 26.4 | **30.4** | 28.4 | 16.8 | 28.4 | 28.0 | 28.4 |
| 20 | 41.2 | 40.8 | **44.0** | 38.0 | 31.2 | 41.2 | 41.2 | 40.0 |
| 27 | 37.2 | 37.6 | 35.6 | 32.4 | 32.4 | 37.2 | 38.0 | **38.8** |
| 31 | **47.6** | **47.6** | 46.4 | 43.2 | 31.6 | **47.6** | **47.6** | 46.4 |
| 50 | 21.2 | 20.8 | 20.8 | 19.6 | 15.6 | 21.2 | 20.8 | 20.8 |
| Avg. | 29.6 | 29.4 | **29.9** | 28.3 | 22.1 | 29.6 | 29.6 | 29.1 |

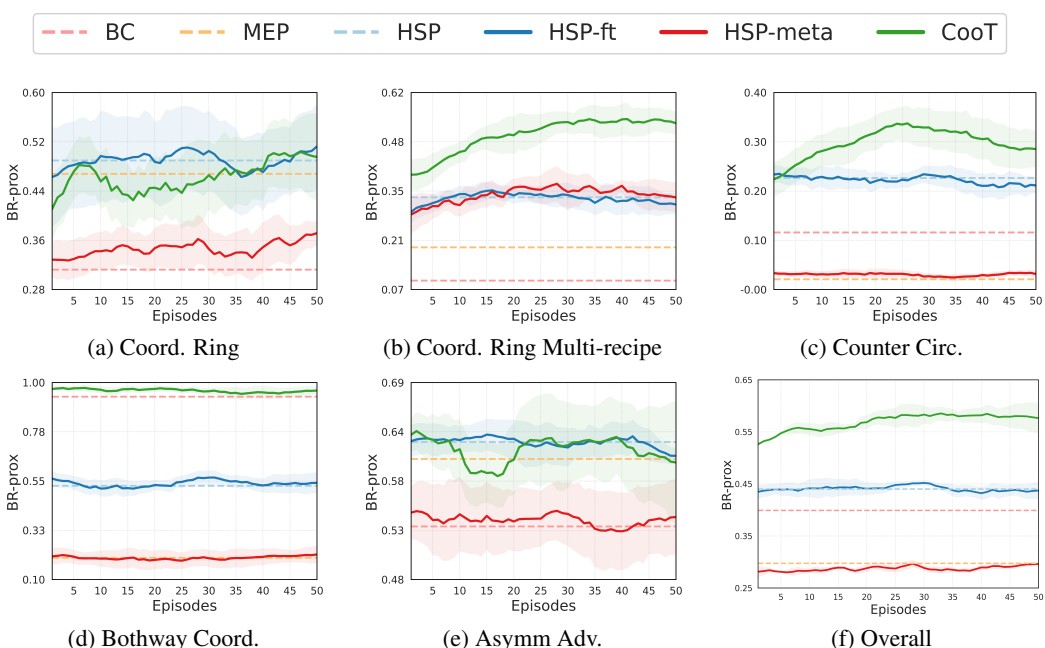

Figure 5: **COOT performance across layouts.** Learning curves on six evaluation layouts: Coord. Ring, Coord. Ring Multi-recipe, Counter Circ., Bothway Coord., Asymm Adv., and the aggregate result across all layouts (Overall).

flat. In *Coord. Ring*, COOT rises from initial BR-prox values around 0.4 to over 0.5, The difference is most striking in the *Coord. Ring Multi-recipe* layout: COOT continues to improve over the full 50 episodes, yet HSP-Meta stagnates early and even falls below the HSP baseline.

These results suggest that the trajectory encoder introduced in HSP-Meta does not produce useful latent representations for coordination. One likely reason is that the reconstruction loss does not align with coordination objectives: subtle behavioral cues are either smoothed out or misrepresented, leaving the policy unable to differentiate between partners. By contrast, COOT conditions directly on observed interactions and learns to map them to best-response actions, leading to consistent adaptation across all layouts.

# D  ADDITIONAL CONTEXTS FOR HUMAN EXPERIMENT

## D.1  EXPERIMENT SETUP

We recruited and verified 36 participants, with a gender distribution of 27 males and 9 females, for the human experiment. Seven participants have prior experience in playing the actual Overcooked!. To mitigate learning effects among the subjects, the order of the agents was randomized. The participants are required to play 200 per episode, 1600 timesteps in total, with 8 episodes for each agent (approximately 5.3 minutes). This leads to a total time of around 30 minutes. The names of the algorithms used by the agents were not visible during the experiments. Agents were differentiated solely by color. Participants were asked to rank the agents after each round, and their trajectories were recorded. All data collection was conducted with the consent of the participants.

## D.2  EXPERIMENT PLATFORM

We built our human evaluation platform on top of the ZSC-Eval benchmark (Wang et al., 2024)[A.1], which provides a standardized environment for testing human-AI coordination in Overcooked. To adapt it to our setting, we modified the system to support repeated interactions with the same agent, enabling context accumulation over multiple episodes.

During the experiment, participants controlled one character using keyboard inputs, while the partner was controlled by one of the four agents under evaluation: COOT, MEP, HSP, or BC. To reduce potential bias, agent identities were hidden and replaced with randomized colors. The interface presented real-time feedback, and all trajectories were automatically recorded for later analysis. Figures 6 to 9 show the platform's interface and experiment flow.

Table 13: **Human feedback comments for different agents.** This table reports representative free-form comments collected from participants during the human–AI evaluation. All comments are anonymized and shown exactly as written, without further editing.

| USER | CooT |
|------|------|
| 1 | It didn't put onion to the pot which already had onion inside. Sometimes it didn't manage to get the onion put on the middle table. |
| 2 | It can understand basic gaming strategy but cannot understand my intention |
| 3 | Do not take onions. More predictable. |
| 4 | better than first |
| 5 | It would block me at first, but it improves collabing with me. I put the onion, and he would take the soup. It would also do my work as well. Generally, it just feels smoother and better along the play. |
| 6 | Disappointed. I found that agent3 is good at the game. However, he refused to adapt to my strategy so waste a lot of time stucking thogather. |
| 7 | agent1 know how to use middle table but often block my way |
| 8 | has some strategies, but some behaviors are meaningless |
| 9 | not very smart |
| 10 | He performs well when playing on his own, so he would take three onions and a plate in the beginning. If I stole his soup, he would freak out and do nothing for a while. However, he walks really fast on his own. |
| 11 | It finds in the begining, I will put onion in the middle table, so it waits for me. Smart! BUT it likes to hold the plate and wait in front of the pot. But this robot is the best to collaborate. |
| 12 | better collaborative, sometime will be at right position |
| 13 | it doesn't put onion into the pot with more onions |
| 14 | A stubbern model. S/he just didn't know how to change the route. However, s/he found my behavior pattern and tried to change his/her behavior. |
| 15 | The agent is smart and sometimes provide hints to me. |
| 16 | often block the road, but seldon hesitate |
| 17 | Agnet4 play well. We play our own rules and get the highest grade. |
| 18 | He keeps take the plate when there is no onion soup ready. |

| 19 | not a speedy start...but I think we'll become a great partner in a near future |
|---|---|
| 20 | so-so |
| 21 | good. |
| 22 | it stop at the first time and it will take the plate at the first time |
| 23 | kinda stupid |
| 24 | Doing well on placing egg, but worse whiling placing dish |
| 25 | hehehe |
| 26 | Pretty good |
| 27 | I think the agent performs best at the beginning. It starts to become confusing after rounds. |
| 28 | Seems not to care much about my play style. Always wanted to achieve its purpose no matter what. Meaning that it would block my way and would not back the fuck off. |
| 29 | The agent can wait me to put onions. |
| 30 | This agent often block my way. We often stood there and look as each other for a long while. It seems that this agent doesn't quite understand maximizing throughput. It usually fill in only one cook and wait there with a plate in hand, leaving the other cook idle. |
| 31 | not helpful |
| 32 | I won't recognize this agent as an AI if I collaborate with it in the real game. As time passed by, the agent tends to be more adative and it won't block my way often. the only point is that he won't put the onion on the oven that already had one or two onion, since it might be more effective and efficient for us to give our meal faster. |
| 33 | At the begining we don't have a good strategy, but we become more efficient and the agent is pretty collaborative |
| 34 | It responded quickly and knowed how to cooperate with me well.. |
| 35 | The agent sometimes block my way. However, it seems to know my playing style and can collabrate with me to some extent. |
| 36 | not bad |

| USER | MEP |
|---|---|
| 1 | it helps me take the onions to the stove. |
| 2 | It can coorperate with me a little bit in the early episode, but not in the later one |
| 3 | Move with initiative. |
| 4 | quite silimilar to first |
| 5 | The performance is not bad. But it gets a little bit worse over time. The agent performs well, but it can't compromise to my plan. |
| 6 | Agent2 kind of understand my strategy at later runs however the as the score gets higher some unseen situation still confuse him. |
| 7 | agent4 can walk around and not stock traffic |
| 8 | Quick guy, but seems to have some prefer pattern, didn't adapt very much |
| 9 | also not very smart |
| 10 | He seems to be thinking about every step I do, and then think about what he should do. This takes up a lot of time and causes inefficiency. Moreover, I think he loves me and likes to block in my way. |
| 11 | I think this robot play the game itself but this one is smart. |
| 12 | Although it play bad, it improve a lot at the end |
| 13 | i expect it to take a plate when i took the third plate but it didn't |
| 14 | S/he is faster than the former one. However. I tried to collaborate with him/her, but nothing happened. |
| 15 | The agent sometimes blocks me but sometimes is clear and follows my strategy. |
| 16 | doesn't follow the same route, block the way sometimes, |
| 17 | Agent1 is smart sometimes but is also stubid sometimes. |
| 18 | He didn't know that he could take the onion from the middle(which I just put there before). |
| 19 | non-improving partner; always work in an averaged level |
| 20 | not good at all |
| 21 | can trace me step but sometimes missed |
| 22 | like a human, but it does not learn my work model |

| | |
|---|---|
| 23 | not the best, but not that bad |
| 24 | Always block my way to go. |
| 25 | hahaha |
| 26 | Co-work good, but not learning |
| 27 | It would help me take the soup, so for the cooperation, i think it performs good, similar to agent 1. |
| 28 | Blocked my path sometimes, but helped me served onion soup ) |
| 29 | Single mode agent |
| 30 | This aggent seems to know how to collaborate. Behaviors such as filling the cook with onions in it first and taking the onions I put on the table are observed. |
| 31 | keep bump into me |
| 32 | I think this guy is performing better and better and he seems to learn how to use the central block. It performs bad at first, too, but it just become better over time. |
| 33 | At the starting rounds, it seems that we developed a good strategy. However, the agent starts to violate the strategy and frequently results in conflict |
| 34 | Sometimes I was blocked by the agent 1. |
| 35 | It peforms well at the begining, so I feel like it can really collaborate with me. However, it didn't seem to adapt to my playing style and seem to became stupid over time. |
| 36 | better |

| USER | HSP |
|---|---|
| 1 | Sometimes it would help me pick up onions but I think we collaborate well. It even put the soup on the table instead of sending it. |
| 2 | It can understand some parts of the rule but lack of the ability to make right decisions, and it can barely coorperate with me. |
| 3 | spin around for no reason. lack sense of the goal. |
| 4 | useless |
| 5 | Not bad. It's slightly self-centered. It can read my intention in some tasks, but it would also do what it wants in some other scenarios. The adaptation over time is not obvious. |
| 6 | very stubben. I use the same strategy every run but agent1 still do stuffs against me. :( |
| 7 | agent3 can only do deliver well |
| 8 | A little bit dumb, seems to know what I try to do sometime, but slow |
| 9 | becomes worse |
| 10 | He seems to be thinking what he should do and what I've done the same time. This causes difficulties in collaboration because human would decide whether to first observe of first perform action, not doing it in the same time. Eventually, he is blocking my way and doing nothing. |
| 11 | this robot find I prefer to put one onion on the table in the middle first and it did it in the last round. Good. BUT it is stupid than first one. |
| 12 | The performance is set betwwen 1 and 4 |
| 13 | doesn't put onion into the pot with two onions in |
| 14 | S/he seems to block my path several times. |
| 15 | The agent always blocks me and do not follow my action. |
| 16 | folow the same route, but delay a little bit |
| 17 | Agnet2's behavier is wired. In the beginning he know how to collaborate with me putting the onion to the stove, but later he forgot and didn't improve. |
| 18 | He is smart but keeps being in front of me. |
| 19 | clumsy guy; I think he's new in kitchen |
| 20 | not good at first, but imporve after that |
| 21 | faster than me |
| 22 | i think better than the green one, but sometime it will stop and does not work. |
| 23 | such a retard |
| 24 | agent seems like walking in circle. |
| 25 | wuwuwu |
| 26 | Pretty bad |
| 27 | It stops moving sometimes and it also blocks my way. i dont think it actually improve over time. |

| 28 | Blocking my path and wouldn't back down, and do not like to serve the onion soup. But over time, it learns to back off when I want to move and serve onion soup. |
|---|---|
| 29 | place onion some where other than oven |
| 30 | This agent seems to be "less confident" on what should do. Hesitations are observed. Path blockage still happens but less frequent than agent 1.. |
| 31 | okay |
| 32 | This AI is quite similar as agent 2, I found it quite confused. Sometimes it respond to my action well sometimes it just put things randomly. Agent 2 often block my way but this guy seems that it is a newbie. Overall the Agent 2 and Agent3 are quite similar. Besides, i have tried to take advantage of the block at the center of the map but both of them can't use it properly. |
| 33 | Sometimes it will block my way and do conflict actions |
| 34 | It seems that agent 2 is a bit srupid that it did'n know how to adjust the order of making dish. Moreover, it often blocked me. |
| 35 | The agent peforms pretty bad, and it didn't adapt to my playing style at all! It often blocked my way and did nothing when I was busy. |
| 36 | bad |

| **USER** | **BC** |
|---|---|
| 1 | It always blocks my way and it would pick up plate when the pot is empty. |
| 2 | It only understand a little bit of the game's rule but being very bad at it |
| 3 | Move with less pattern. |
| 4 | great |
| 5 | The agent is more self centered. It can't really adapt to my behavior and read my intentions. Although there are some improvements along the way, yet it gets worse in the end. |
| 6 | Nice guy. |
| 7 | agent2 often do something weird |
| 8 | has diverse behaviors, but some are meaningless |
| 9 | such an idiot, keep blocking my way |
| 10 | Agent1 performs well when I do new moves like take an onion, but performs bad on deciding actions with the oven. |
| 11 | Although in the beginning, this robot did nothing useful for example, it prefer to put the onion in the different pot and hold the plate in front of an unfill pot, in the last round it figure out I would like to put onion in the middle table therefore, it will wait for onion near the pot. |
| 12 | Not very collaborative |
| 13 | sometimes block my way but not always |
| 14 | S/he didn't block my path and utilized the middle table. |
| 15 | The agent always blocks my strategy and takes the wrong items. It doesn't collaborate with me well. |
| 16 | make the wrong step often |
| 17 | Agent3 can learn how to use the middle area. However, I don't know wht it likes to take the plate when nothing is cooking. |
| 18 | He acts stupid at first but learns very fast. |
| 19 | a very speedy partner; I think I didn't leverage his strategy of putting stuff on the middle island |
| 20 | better than agent1 |
| 21 | bad |
| 22 | sometime the derection for agent1 will be different for me. |
| 23 | the smartest one |
| 24 | always stand on the cooking area. |
| 25 | yayaya |
| 26 | DOESN'T IMPROVE |
| 27 | It sometimes blocked my way, for the adaptive ability, i think it performs similarly. |
| 28 | Very bad player. At first, would help to change from onion to plate to serve. But over time, it became brain damaged and would block my way and would not do anything. |
| 29 | The agent does not know how to use plates. |

| 30 | It seems that we are working separately. This agent has neither help me nor undo my work.It seems to adapt a little bit but not apparent. |
| 31 | keep learning but not really good |
| 32 | It is more stupid and i can easily distinguidh that it is an AI. Sometimes it did weird behavior, sometimes it just idle and didn't do anything. In the round 7 the AI guy just stop in front of the ai for about 15 mins, while both ovens are filled with soup. I can't do anything then. I didn't see that he improve by any means. It sometimes block my way, more often then agent 1. |
| 33 | Very bad at collaboration. Can hardly develop a playing strategy. Seems to be playing by itself |
| 34 | It responded quickly. |
| 35 | The agents often block my way. It's not really helpful. |
| 36 | Very bad |

### D.3 QUALITATIVE FEEDBACK FROM PARTICIPANTS

To complement the quantitative rankings, we also collected qualitative feedback from participants after each interaction session. These open-ended comments provide insight into subjective impressions of each agent's behavior, including adaptiveness, blocking tendencies, and responsiveness. We organize these comments in Table 13, aligned with the BC, MEP, HSP, COOT order based on participant rankings.

To validate that the observed advantages were behaviorally grounded and not attributable to a halo effect, we conducted a qualitative analysis of all 144 participant responses. This manual coding process categorized comments along two independent dimensions: Explicit Adaptation, defined as evidence of progression or successful strategy modification using the method; and Negative Behaviors, which analyzed mentions of actions such as blocking and obstruction. To ensure mutual exclusivity, responses coded for Explicit Adaptation were exempt from inclusion in the Negative Behaviors dimension, reflecting the difference between demonstrating successful strategy use and focusing solely on obstacles and failures within the qualitative data. The resulting distribution of codes, which is itemized by participant ID in Table 14, supports the findings derived from the quantitative rankings.

Table 14: **Qualitative coding distribution.** N is the number of comments.

(a) **Negative Behaviors**

| Agent | N | Participant IDs |
|---|---|---|
| BC | 19 | 1, 2, 3, 5, 6, 9, 12, 13, 15, 16, 21, 24, 26, 28, 29, 32, 33, 35, 36 |
| MEP | 11 | 9, 10, 11, 14, 16, 18, 19, 20, 24, 31, 34 |
| HSP | 21 | 2, 3, 4, 6, 8, 10, 14, 15, 17, 20, 22, 23, 24, 26, 27, 31, 32, 33, 34, 35, 36 |
| CooT | 12 | 1, 2, 7, 8, 9, 10, 14, 22, 23, 28, 30, 31, 35 |

(b) **Explicit Adaptation**

| Agent | N | Participant IDs |
|---|---|---|
| BC | 3 | 11, 18, 31 |
| MEP | 4 | 6, 21, 30, 32 |
| HSP | 5 | 5, 8, 11, 20, 28 |
| CooT | 9 | 5, 11, 12, 14, 29, 32, 33, 34, 35 |

| Layout | Game Length (sec) | Game number | Agent ID |
|--------|-------------------|-------------|----------|
| random1 | 400 | 3 of 4 | 3 of 4 |

When playing, please make sure to pay attention to the hat colors of your partners!
You will be asked to rank the agents (identified by their hat colors) according to their performance.

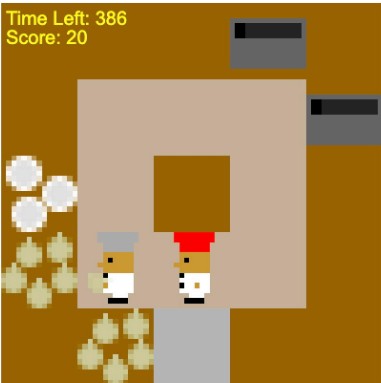

Figure 6: **Main experiment layout for human study.** *Coord. Ring* is chosen since it requires both navigation and ingredient coordination. Thus, it can observe more coordination situations.

## Questionnaire

Please rank the agents by dragging the corresponding figures based on your feelings of the agents' cooperation ability.
Please rank the agents **from best to worst, from top to bottom**.

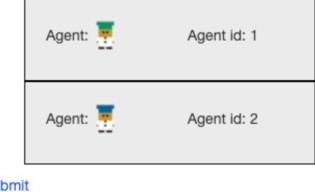

Submit

Figure 7: **Human subjective perception ranking system.** Interface used in the user study for collecting participants' subjective rankings of agents. After each round, participants were asked to compare all agents they interacted with and select which partner they preferred most. This ranking procedure complements quantitative metrics by capturing human impressions of collaboration quality.

## Experimental Statement

### 1. Purpose

You have been asked to participate in a research study that studies human-AI coordination. We would like your permission to enroll you as a participant in this research study.

The instruments involved in the experiment are a computer screen and a keyboard. The experimental task consisted of playing the computer game Overcooked and manipulating the keyboard to coordinate with the AI agent to cook and serve dishes.

### 2. Procedure

In this study, you should read the experimental instructions and ensure that you understand the experimental content. The whole experiment process lasts about **30** minutes, and the experiment is divided into the following steps:

(1) Read and sign the experimental statement, and you need to fill in a questionnaire ;

(2) Test the experimental instrument, and adjust the seat height, sitting posture, and the distance between your eyes and the screen. Please ensure that you are in a comfortable sitting position during the experiment ;

(3) You will first try out the game actions you learned in the tutorial within a simple layout to familiarize yourself with the game mechanics;

(4) Start the formal experiment. Please cooperate with the AI agents to get as much scores as possible. You will play with 4 agents in 1 layout. You need to rank the performance of these four agents. After each round, we will ask you to add the current agent to the ranking. After the game ends in each layout, we need to confirm your ranking of the agents.

### 3. Risks and Discomforts

The only potential risk factor for this experiment is trace electron radiation from the computer. Relevant studies have shown that radiation from computers and related peripherals will not cause harm to the human body.

### 4. Compensation

Each participant who completes the experiment will be paid around 6~7 USD.

### 5. Confidentiality

The results of this study may be published in an academic journal/book or used for teaching purposes. However, your name or other identifiers will not be used in any publication or teaching materials without your specific permission. In addition, if photographs, audio tapes or videotapes were taken during the study that would identify you, then you must give special permission for their use.

I confirm that the purpose of the research, the study procedures and the possible risks and discomforts as well as potential benefits that I may experience have been explained to me. All my questions have been satisfactorily answered. I have read this consent form. Clicking the button below indicates my willingness to participate in this study.

Figure 8: **Statements for human study.** Consent and instruction form provided to participants before the experiment. It outlines the purpose of the study (human–AI coordination in Overcooked), the procedure (tutorial, gameplay with four agents, and post-round rankings), potential risks and discomforts, compensation details, and confidentiality terms. This ensured that participants were fully informed and agreed to the study protocol before beginning the human–agent collaboration tasks.

## Instructions

Please read the following instructions carefully.

In this task, you will play in a cooking game as one of the two chefs in a restaurant that serves onion soup. The chef in you control wearing a gray hat.

One of the game layouts looks like:

There are a number of objects in the game, labeled here:

**Movement and interactions**

You can move up, down, left, and right using the **arrow keys**, and interact with objects using the **spacebar**.

You can interact with objects by facing them and pressing **space bar**.

Note that you and your partner **cannot occupy the same location**.

**Cooking**

Cooking Soup    Cooked Soup

Once 3 onions are in the pot, the soup begins to cook. After the timer gets to 20, the soup will be ready to be served. To serve the soup, bring a dish over and interact with the pot.

**Goal**

Your goal in this task is to serve as many of the orders as you can before each level ends. The current score and time left for you are shown in the upper left of game.

After clicking "Start Playing", you will first play in a warmup trial, where scores will not be recorded.

After the warmup trial, the official experiments will be conducted in 3 layouts. You will complete 7 games with 7 different agents in each layout.

When playing, please make sure to pay attention to the **hat colors** of your partners! You will be asked to rank the agents (identified by their hat colors) according to their performance.

Figure 9: **Instructions for human study.** Guidelines were presented to participants before starting the Overcooked sessions. The instructions described the objective of the task, the number of episodes to be played, and the anonymity of the partner agents (displayed only by color). This ensured participants understood the procedure while minimizing bias toward specific algorithms.

Table 15: **Goal score on Google Research Football.** We report the goal rate ($\uparrow$), defined as the number of goals achieved within the 200-step evaluation horizon. Results compare CooT, HSP, and MEP against 10 evaluation partners selected for maximum behavioral diversity (Appendix B.6). Each method is trained with three seeds, and each trained model is evaluated with 50 rollouts. The reported mean $\pm$ std is computed from the per-seed averages. Higher scores indicate stronger coordinated play.

| Method | Goal Rate |
|---|---|
| MEP | 0.75±0.12 |
| HSP | 0.90±0.10 |
| CooT (Ours) | **1.35±0.26** |

## E    COORDINATION BEYOND OVERCOOKED: GOOGLE RESEARCH FOOTBALL

The previous sections focus on Overcooked, a compact two-agent domain that highlights rapid adaptation and robustness to partner shifts. To examine whether COOT 's advantages extend to more complex multi-agent settings, we benchmark it against HSP and MEP, the strongest Overcooked baselines, in Google Research Football (GRF) (Kurach et al., 2020), using the ZSC-Eval evaluation protocol (Wang et al., 2024). HSP and MEP are trained with 10M online RL steps, whereas COOT is trained purely from a 5M offline dataset, making the comparison conservative in favor of the baselines (further details in Appendix 6 and Table 7).

**Environments.** We utilize the "3 vs 1 with Keeper" scenario from GRF. In this setting, three cooperative offensive agents work together to score against one defender and a goalkeeper. We denote the three attackers as $P_0$, $P_1$, and $P_2$, which allows us to specify their roles and behavior preferences precisely (see Appendix A.2 for details).

**Behavior-preferring agents and best response.** We define two sets of discrete events for $P_0$ and $P_2$, and generate 144 biased reward functions through linear combinations of these events. Using these rewards, we train $P_0$ and $P_2$ jointly with $P_1$, producing diverse partner behaviors. After training, we collect the trajectories of $P_1$ acting as the best response to each pair of partners. These best-response trajectories serve as the training data for CooT.

**Setup and metrics.** In our configuration, each episode lasts up to 200 steps and resets when a goal is scored or when the environment terminates prematurely (e.g., due to out-of-bounds or a change in ball possession). Agent performance is measured by the goal rate, defined as the number of successful goals achieved within each episode. Our implementation closely follows the GRF integration in ZSC-Eval. Specifically, the evaluation partners are constructed using two components: the hidden-utility biased rewards we design (as detailed in Table 16), and the MAPPO training procedure employed in ZSC-Eval's GRF module. We then select a behaviorally diverse evaluation subset of size 10 using feature embeddings and determinantal point process sampling (further details in Appendix B.6).

**Results.** GRF presents a substantially larger and more continuous coordination space than Overcooked, and COOT achieves competitive goal scores across diverse evaluation partners (Table 15). Consistent with our observations in Overcooked, COOT 's advantage becomes more pronounced when the coordination demands increase: tasks that require multi-agent spacing, multi-step passing, or maintaining offensive formations benefit from its ability to absorb interaction histories and quickly infer partner tendencies. These results indicate that CooT's in-context adaptation mechanism generalizes beyond structured grid-based settings and remains effective in high-coordination, continuous multi-agent environments.

## F    EXTENDED RELATED WORK

**Opponent Modeling.** The development of competitive agents in multi-agent scenarios, especially against unknown and nonstationary opponents, presents a significant challenge. One effective strategy for addressing this is to equip the agent with the ability to model its opponent. This approach, known as opponent modeling, involves conditioning the agent's policy not only on its environment

Table 16: **Events and biased reward under different layouts.** Each entry lists all possible biased reward values that can occur for the corresponding event under the given layout type.

| Events | $P_0$ | $P_2$ |
|---|---|---|
| pass | -1,1 | 0 |
| shot | 0 | -1,1 |
| possession | -0.1,0,0.1 | -0.1,0,0.1 |
| score | 1,5 | 1,5 |

observation but also on predictions about relevant properties of the opponent, such as their policies and goals. Greedy when Sure and Conservative when Uncertain about the Opponents (GSCU) (Fu et al., 2022) solves this by selecting between a real-time greedy policy and a fixed conservative policy using an adversarial bandit algorithm.

**Social Dilemmas.** The concepts of cooperation and competition are fundamental to the study of social systems in both nature and artificial intelligence. Multi-agent reinforcement learning (MARL) has achieved notable success in settings like Go and Starcraft, which are complex but typically fixed-team and zero-sum. However, the real world often involves mixed-motive interactions that are neither purely zero-sum nor defined by fixed teams. In these settings, agents constantly face social dilemmas, where their individual interests conflict with the collective well-being of the group. Randomized Uncertain Social Preferences (RUSP) (Baker, 2020) solves this by expanding the distribution of environments the agents are trained in, specifically by introducing randomized, uncertain, and asymmetric prosocial preferences. This novel environment augmentation pressures agents to learn socially reactive policies, such as reciprocity and team formation, which are necessary for cooperation.

## G ADITIONAL BASELINE

We acknowledge that several context-based meta-RL approaches could be applied to the ad-hoc teamwork (AHT) setting. Using only HSP-meta to represent this series of methods may therefore be incomplete. To strengthen our empirical results, we additionally adapt Fast Peer Adaptation with Context-aware Exploration (PACE) (Ma et al., 2024) into our cooperative setting. The resulting baseline, which we call HSP-PACE, enables us to more directly compare CooT against a representative context-based approach designed for fast adaptation from recent interaction history.

**Method Overview.** PACE is a context-aware meta-RL method designed to identify and adapt to different peers in multi-agent environments. It uses a context encoder to summarize recent interaction episodes into a latent context, similar to the encoder used in our HSP-meta baseline. On top of this encoding, a peer classifier predicts the identity or type of the peer and produces an intrinsic exploration reward based on the posterior probabilities of the actual peer agents. The classifier is further trained with an auxiliary loss between the predicted peer distribution and the true peer ID. By conditioning the policy on the inferred peer embedding and using uncertainty-driven intrinsic rewards to encourage informative interactions, PACE enables fast adaptation across diverse peer strategies.

**Adapting PACE into HSP.** To construct a fair comparison, we integrate PACE's core mechanisms into HSP-meta, resulting in HSP-PACE. Specifically, we add a peer-identifier trained to classify partner policies from the latent context produced by HSP-meta's encoder. The identifier produces a posterior distribution over partner IDs, which we use following PACE's design in two ways: (i) as the target for an auxiliary classification loss, and (ii) to compute an intrinsic exploration reward that encourages the agent to collect interactions that help disambiguate partner identities. During training, this intrinsic reward is combined with the environment reward; at test time, only the latent context (without the exploration reward) conditions the policy. These modifications preserve PACE's identity classification and uncertainty-driven exploration mechanisms while adapting them to the AHT setting.

**Results.** As shown in Table 17, HSP-PACE demonstrates highly inconsistent performance that collapses under increased task complexity. In the less demanding Coord. Ring layout, HSP-PACE achieves a moderate reward of 33.94, outperforming HSP-meta (29.84). This suggests that in simple

Table 17: **Additional baseline: HSP-PACE (added during rebuttal).** We include HSP-PACE as an extra baseline following the reviewer's suggestion. HSP-PACE performs slightly above HSP-meta on Coord. Ring but remains below the strongest baselines (MEP, HSP/HSP-ft) and COOT. On the other hand, HSP-PACE suffers a significant failure in the Coord. Ring Multi-recipe layout. In this demanding environment, HSP-PACE records the lowest mean reward of 4.43 with a high standard deviation ($\pm$9.02), indicating a fundamental breakdown of the coordination mechanism. All results report mean $\pm$ std over 3 training seeds, with 50 evaluation rollouts per seed.

| Layout | Coord. Ring | | Coord. Ring Multi-recipe | |
| --- | --- | --- | --- | --- |
| | Reward | BR-prox | Reward | BR-prox |
| BC | 26.24$\pm$1.80 | 0.31$\pm$0.02 | 8.97$\pm$0.49 | 0.10$\pm$0.01 |
| MEP | **40.30**$\pm$**3.45** | **0.47**$\pm$**0.04** | 16.64$\pm$1.16 | 0.19$\pm$0.02 |
| HSP | **41.10**$\pm$10.03 | **0.49**$\pm$0.10 | 29.35$\pm$3.77 | 0.33$\pm$0.04 |
| HSP-ft | **41.30**$\pm$9.85 | **0.49**$\pm$0.10 | 29.24$\pm$3.75 | 0.33$\pm$0.04 |
| HSP-meta | 29.84$\pm$3.92 | 0.35$\pm$0.04 | 30.21$\pm$1.37 | 0.34$\pm$0.02 |
| HSP-PACE | 33.94$\pm$3.21 | 0.40$\pm$0.03 | 4.43$\pm$9.02 | 0.05$\pm$0.10 |
| COOT (Ours) | **38.30**$\pm$3.71 | **0.47**$\pm$0.06 | **45.96**$\pm$**3.99** | **0.50**$\pm$**0.04** |

layouts, the method can achieve a baseline level of coordination, although it still lags behind the top-performing methods like CooT (38.30) and other HSP variants ($\approx$41.0). In the complex Coord. Ring Multi-recipe layout, HSP-PACE achieves an extremely low reward of 4.43.

**Analysis.** The failure of HSP-PACE in Coord. Ring Multi-recipe might come from two compounding issues: policy-pool mismatch and the intrinsic exploration reward. In PACE's original PO-Overcooked environment, agents have access to several ingredients and recipes, and the "diverse" policy pool is constructed by assigning each peer an ingredient-oriented preference, which makes the peer agent ignore everything other than the target ingredients. These non-overlapping preferences make partners cleanly distinguishable, which fits the design of PACE's peer-identifier. In our setting, however, biased partners are trained with an RL algorithm under biased rewards rather than explicit behavior restrictions, so partners do not completely avoid non-preferred behaviors. This difference makes peer identification substantially harder and could cause the classifier to overfit. This weakness is critically exposed in the complex environment, where the demand for high-precision, collision-free coordination is high. Furthermore, PACE's intrinsic exploration reward may encourage the agent to deviate from cooperative behavior to gather identity-revealing signals, which is counterproductive under fixed training timesteps. As a result, HSP-PACE fails to stabilize an effective cooperative policy and is outperformed by other context-based methods like HSP-meta and CooT.

## H ALGORITHM

---

**Algorithm 1** Agent Pool Construction and Training

---

1: // Generating agent pool
2: Initialize empty pool $\Pi_0$
3: **for** $i$ in $P$ **do**
4:     Sample hidden reward function $r_i^w$ from reward space $R$
5:     Train $\pi_i^{\mathrm{p}}$ and its best response $\pi_i^{\mathrm{br}}$ using PPO
6:     Add $(\pi_i^{\mathrm{p}}, \pi_i^{\mathrm{br}})$ to $\Pi_0$
7: **end for**
8: // Partner selection for training
9: Initialize empty training pool $\Pi_{train}$
10: **for** $(\pi_i^{\mathrm{p}}, \pi_i^{\mathrm{br}})$ in $\Pi_0$ **do**
11:     Rollout trajectories and Compute event-based diversity $d_i$ of $\pi_i^{\mathrm{br}}$
12: **end for**
13: Select top-M agents with highest $d_i$ values as $\mathcal{S}$
14: Add corresponding agents to $\Pi_{train}$
15: Remove corresponding agenst from $\Pi_0$
16: // Construct training dataset
17: Initialize empty dataset $D$
18: **for** $(\pi_j^{\mathrm{p}}, \pi_j^{\mathrm{br}})$ in $\Pi_{train}$ **do**
19:     **for** $k$ in $K$ **do**
20:         Rollout $T$ trajectories as context $C$
21:         **for** $l$ in $L$ **do**
22:             Sample query state $s_h \sim C$
23:             Let $a^\star = \pi_j^{\mathrm{br}}(s_h)$
24:             Add data $(s_h, C, a^\star)$ to $D$
25:         **end for**
26:     **end for**
27: **end for**
28: // Model training
29: Initialize model $M_\theta$
30: **while** not converged **do**
31:     Sample $(s_h, \tau_j, a^\star)$ from $D$
32:     Predict action distribution $\hat{p} = M_\theta(\cdot \mid s_h, \tau_j)$
33:     Compute loss $\mathcal{L}$ given $\hat{p}$ and Update $\theta$
34: **end while**

---

**Algorithm 2** Evaluation and Online Deployment (Wang et al., 2024)

---

1: // Evaluation partner selection
2: **for** $(\pi_i^{\mathrm{p}}, \pi_i^{\mathrm{br}})$ in $\Pi_0$ **do**
3:     Rollout trajectories $\tau_i$
4:     Embed features of $\tau_i$ into $\phi_i$
5: **end for**
6: Compute similarity matrix $\mathbf{K}$ from $\{\phi_i\}_{i=1}^K$
7: Sample subset $\mathcal{S}$ from top-N candidates of $\mathbf{K}$
8: Define evaluation set $\Pi_{\mathrm{eval}} = \{\pi_s^{\mathrm{p}}\}_{s \in \mathcal{S}}$
9: // Online deployment
10: Sample unseen partner $\pi_s^{\mathrm{p}} \sim \Pi_{\mathrm{eval}}$
11: Initialize fixed-length context $C = \{\}$
12: **for** episode $= 1$ to $E$ **do**
13:     **for** timestep $t = 1$ to $Z$ **do**
14:         Observe $s_t$, predict $a_t \sim M_\theta(\cdot \mid s_t, C)$
15:         Execute $a_t$ with partner, observe $(s_t, a_t, r_t)$
16:     **end for**
17:     Append episode trajectory to context $C$
18: **end for**

---

## I THE USE OF LARGE LANGUAGE MODELS

We used large language models (LLMs) in limited ways that did not affect the scientific contributions of this work. Specifically, LLMs were employed to (1) polish and improve the clarity of writing without altering the technical content, (2) help organize and summarize qualitative feedback collected from human study participants, and (3) assist in designing and refining figures for presentation purposes. All conceptual, methodological, and analytical contributions, including study design, data analysis, and interpretation of results, were carried out solely by the authors.

