# OpenReview forum: "CooT: Learning to Coordinate In-Context with Coordination Transformers"
_ICLR.cc/2026/Conference — Submitted to ICLR 2026_

### Official Review · Reviewer_Krp4 · 2025-10-27

**Soundness:** 3
**Presentation:** 3
**Contribution:** 2
**Rating:** 2
**Confidence:** 3

**Summary:**

The paper introduces COOT, a novel in-context learning framework designed to enable agents to coordinate with previously unseen partners by conditioning their actions on recent interaction histories. COOT is trained on trajectories from diverse pairs of agents whose behavior is driven by hidden reward functions, framing coordination as a Hidden-Utility Markov Game (HU-MG) problem. In experiments using Overcooked, COOT consistently outperforms strong baselines (including population-based and gradient-based methods) across diverse coordination tasks, achieving stable and rapid adaptation without any parameter updates. Human evaluations also ranked COOT as the most effective and preferred collaborator, demonstrating its robustness against the variability of human partners.

**Strengths:**

- COOT consistently outperforms strong population-based baselines (HSP, MEP) across diverse, coordination-heavy Overcooked layouts, including the challenging multi-recipe variant.
- The user study provides strong evidence that COOT's advantage extends to real-world variability.
- The controlled partner-swap experiment was a creative way of demonstrating COOT’s capability to recalibrate its strategy to an abruptly changing partner

**Weaknesses:**

- If the partner you’re interacting with has a different hidden reward function then this is effectively a different MDP since they’ll take different actions making the ego agent’s transition dynamics different. Generating many diverse trajectories and training on them then seems exactly similar to [RUSP](https://arxiv.org/abs/2011.05373) or [CEC](https://arxiv.org/abs/2504.12714). The main novelty seems to come from using a transformer compared to LSTMs which that work uses, but even here that doesn’t seem like too big of an architectural contribution, and there’s no comparison to these models.
- It is unclear whether this method addresses the zero shot or few shot coordination case, it seems to frame the method in the latter but the details of how it evaluates it in the former for both AI-AI comparison and Human-AI comparisons are not clear
- For the qualitative metrics, did participants say anything about the other models that were similar to what they said about Coot? If so then this isn’t a strong signal about peoples’ experience with the models.
- The computational costs of this method are not well addressed, despite efficiency being a core claim of this method
- Only Overcooked is used as a baseline

**Questions:**

- The training methodology—generating diverse trajectories from agents with varying hidden reward functions and training a context-conditioned policy—appears functionally similar to approaches like RUSP and CEC. Given that the main architectural difference is the use of a Transformer over an LSTM, can the authors perform a direct quantitative comparison to those methods which also claimed SOTA?
- Can the authors clarify whether this is a few-shot or zero-shot method and what the evaluation settings were done in?
- Can the authors please provide a more granular breakdown of the qualitative feedback (e.g., coding and quantifying the frequency of terms like "adapts" vs. "blocks") to demonstrate that COOT’s perceived superiority is statistically or subjectively unique, rather than just the result of general human positivity towards the best-performing agent.
- Given the complexity of the Transformer architecture, can the authors please provide a comprehensive analysis of the full computational cost, including the total memory footprint and the amortized cost of pre-training the expert best-response policies used to generate the large offline dataset? Does the initial cost of generating this dataset outweigh the fine-tuning cost of the baselines?
- The entire evaluation is confined to the Overcooked environment. Given that coordination challenges are fundamentally domain-dependent, how do the authors think about the general applicability of COOT without demonstrating performance on a second, structurally different multi-agent coordination benchmark (e.g., a capture-the-flag setting, or a mixed-motive social dilemma)?

---

> ### Author Response · Authors · 2025-11-22
> **Rebuttal of Authors**
>
> # Response to Reviewer Krp4 [1/2]
> We thank the reviewer for the thoughtful feedback. Your questions about our method’s novelty, evaluation setup, human-study details, computational cost, and broader generalizability are all appreciated. Please find our responses below.
>
> > CooT appears functionally similar to approaches like RUSP and CEC.
>
> We agree that RUSP [1] and CEC [2] share some conceptual similarities with our work. However, their actual goals and behaviors are quite different from what CooT is designed to do.
>
> RUSP uses random-preference agents to augment the training environment so that policies learn general social behaviors like reciprocity, reputation, and team formation. Its history encoder, implemented as an LSTM, simply compresses environment states into compact latent representations. While CooT trains the transformer on the trajectories of best-response policies allowing the model to read a few trajectories and adjust its behavior as a best response during the inference.
>
> CEC focuses on cross-environment coordination rather than partner adaptation, which is the core objective of our work. Although the CEC paper mentions that cross-environment training may help partner generalization, its human-study results did not provide strong evidence for this claim. Moreover, CEC relies on diverse procedurally generated layouts, while our environment does not support random layout generation, making a faithful implementation infeasible.
>
> Because of these differences in goals and required settings, adding RUSP or CEC as baselines would not give a fair or useful comparison. We have instead added discussions about those methods in our revised manuscript to contextualize our approach and highlight the key distinctions (see Chapter 2 and Appendix F).
>
> > Whether this is a few-shot or zero-shot method and what the evaluation settings were done in.
>
> CooT is a few-shot coordination method.
> This follows directly from its in-context learning mechanism as additional interaction trajectories enter the context buffer, the model conditions its action predictions on these behaviorally relevant examples, allowing it to better align with the best-response patterns present in the offline dataset. However, unlike the common understanding of few-shot learning methods, CooT learns to coordiate in a few-shot manner without parameter updates.
>
> Accordingly, our evaluation, described in Sec. 5.1 and Sec. 5.3, is conducted in a few-shot setting: 50 episodes for AI–AI comparison and 8 episodes for human–AI interaction (reduced to avoid fatigue), both of which allow the context buffer to evolve over time. This setup aligns with CooT’s intended usage, where adaptation arises from conditioning on an accumulated interaction history rather than from gradient-based fine-tuning.
>
> > A more granular breakdown of the qualitative feedback.
>
> Thank you for raising this question. We agree that a more granular analysis can help confirm that CooT’s subjective advantage is not merely due to general positivity toward the best-performing agent. While simple keyword counting (e.g., raw frequency of “adapt” or “block”) can be misleading, because many comments contain mixed or context-dependent statements, we performed a manual qualitative coding of all 144 responses, supported by a simple tool-assisted pass for organizing the textual data, and it was double-checked by human.
> Following standard qualitative-analysis practice, each comment was coded along two dimensions: explicit adaptation and negative behaviors (blocking, obstruction, and more).
> - CooT has the **highest rate of explicit adaptation mentions (9/36, 25%)** over MEP (4/36, 11.1%), HSP (4/36, 11.1%), and BC (3/36, 8.3%).
> - CooT (12/36, 36.1%) also has a **comparable rate of negative comments** with MEP (12/36, 33.3%), lower than BC (19/36, 52.8%) and HSP (21/36, 58.3%)
>
> The coded comments and adaptivity/collaboration ratings (Table 3) both indicate that CooT’s perceived advantage is behaviorally grounded: participants attribute adaptation to CooT far more often, and quantitative ratings likewise rank it highest. We believe this additional analysis addresses the concern regarding the source of CooT's advantage, and we would be happy to engage in further discussion regarding this analysis. We have now included these results in Tables 13 and 14, and added an extended analysis in Appendix D.3.

---

> > ### Author Response · Authors · 2025-11-22
> > **Rebuttal of Authors**
> >
> > # Response to Reviewer Krp4 [2/2]
> > > Comprehensive analysis of the full computational cost, including the total memory footprint and the amortized cost of pre-training the expert best-response policies.
> >
> > > Does the initial cost of generating this dataset outweigh the fine-tuning cost of the baselines?
> >
> > The amortized cost of pre-trained policy pool is not unique to our method. All population-based baselines (e.g., HSP, MEP) require training partner policies or diverse policy pools at a similar scale.
> >
> > Transformer pre-training is fully offline. As shown in Appendix B.4, it required only ~19 GPU-hours on a single 3090 Ti. The computational cost of training CooT is modest. While population-based and meta-RL-based baselines (e.g., MEP, HSP) require **per-partner adaptation** via online training. In our setup, this takes ~5 hours, even with 100 parallel environment threads, and must be repeated for each new teammate or configuration.
> >
> > As for the HSP-ft variant in Table 1, it was fine-tuned for only 50 episodes to illustrate the difference between gradient-based adaptation and CooT’s in-context coordination, so the cost is negligible and included only to highlight the contrast with CooT’s gradient-free in-context adaptation.
> >
> > The only cost that affects deployment is inference latency, where CooT remains efficient: 2.41 ms per step vs. 1.75 ms for MEP/HSP (Table 8), well within real-time coordination limits.
> >
> > Overall, CooT’s computation profile only requires a single offline pre-training stage, after which the model is reused without per-partner adaptation, whereas online baselines must perform cross-play or repeat fine-tuning for every new teammate. At inference time, CooT remains lightweight and competitive. We have added more relevant computational details in Appendix B.4 and B.5.
> >
> > > How do the authors think about the general applicability of COOT without demonstrating performance on a second benchmark?
> >
> > We agree that coordination challenges vary significantly across domains, and whether CooT applies depends on the structure of incentives and the meaning of “best response” in each environment.
> >
> > Overcooked is a fully cooperative domain with a clear notion of biased partners: partners may act according to hidden preferences that are misaligned with the explicit team objective.
> >
> > In contrast, agents in mixed-motive social dilemmas are not acting as a team. Here, responding optimally to another agent is closer to opponent modeling than team coordination, and the meaning of “best response” does not translate into cooperative benefit. For this reason, social-dilemma settings are misaligned with the problem CooT is intended to solve.
> >
> > Capture-the-Flag, however, is more compatible with our framework. Although competitive elements exist, teammates often adopt different strategies or roles, and a best-response to a teammate’s strategies can meaningfully improve coordination. In principle, CooT can generalize to this type of domain by training on diverse best-response agents, and this matches the kind of adaptation ability we demonstrate in our paper.
> >
> > Furthermore, to test CooT's generality beyond the Overcooked domain, we have been running experiments in **Google Research Football (GRF)** [3], a setting that differs from Overcooked in several important ways:
> > - **Multi-agent team structure** (3 vs. 1 setting), increasing coordination complexity compared to Overcooked tasks.
> > - **Larger and more diverse action space**, resulting in a broader range of partner behaviors and coordination styles.
> >
> > These properties make GRF a compelling complement to Overcooked for testing cooperative generalization.
> > We are committed to submitting the completed results and discussion well in advance of the author comment deadline.
> >
> > [1] Baker, Bowen. "Emergent reciprocity and team formation from randomized uncertain social preferences." Advances in neural information processing systems 33 (2020): 15786-15799.
> >
> > [2] Jha, Kunal, et al. "Cross-environment Cooperation Enables Zero-shot Multi-agent Coordination." arXiv preprint arXiv:2504.12714 (2025).
> >
> > [3] Kurach, Karol, et al. "Google research football: A novel reinforcement learning environment." Proceedings of the AAAI conference on artificial intelligence. Vol. 34. No. 04. 2020.

---

> > > ### Comment · Reviewer_Krp4 · 2025-11-26
> > >
> > > - Thank you for clarifying the distinction between CEC and CooT. For RUSP, however, I'm still not sure why the authors feel this is not a fair point of comparison? If the learned policy supports general behaviors, presumably this includes cooperation, making the key distinction the choice of LSTM vs Transformer?
> > >
> > > - "8 episodes for human-AI interaction" - do the authors mean that the same agent played a human for 8 episodes, or that it saw the human play 7 episodes potentially with other agents and then used those trajectories as context when collaborating on the 8th.
> > >
> > > - Thank you for the qualitative feedback breakdown, this was insightful. How did the tool-assisted pass work?
> > >
> > > - Thank you for running experiments on additional environments. Please keep me posted on the results when they come in.

---

> > > > ### Author Response · Authors · 2025-12-03
> > > > **Further Response to Reviewer Krp4 [1/2]**
> > > >
> > > > > For RUSP, however, I'm still not sure why the authors feel this is not a fair point of comparison? If the learned policy supports general behaviors, presumably this includes cooperation, making the key distinction the choice of LSTM vs Transformer?
> > > >
> > > > Thank you once again for seeking deeper clarity on our decision to treat RUSP as a related but non-comparable method. We recognize that the term "general behavior" can be misleading, and we must clarify that the foundations of the two works make a direct comparison unfeasible.
> > > >
> > > > **(1) Different definitions of “general behavior.”**
> > > > RUSP targets social dilemmas where a partner may cooperate or act adversarially, emphasizing responses to uncooperative partners and when (and with whom) to form temporary alliances, so its “general behaviors” are specific to mixed-motive settings.
> > > >
> > > > Our ad-hoc teamwork setting assumes fully cooperative partners; the challenge is to infer a partner’s style and coordinate smoothly.
> > > > These notions of “general behavior” are fundamentally different.
> > > >
> > > > **(2) RUSP depends on reward transformation, which is not feasible in our setting**
> > > > RUSP uses a reward transformation matrix and agent-specific payoff uncertainty to modify training incentives, making partners appear helpful or adversarial, mechanisms essential for social-dilemma settings where betrayal is possible. This kind of reward transformation is not feasible in Overcooked/GRF, which assumes fully cooperative partners and a shared team objective; changing the reward would define a different problem, not a comparable ad-hoc teamwork baseline. (For clarity: the event-based hidden reward is used only to generate diverse partner policies when collecting best-response rollouts and is never used to train CooT.)
> > > >
> > > > **(3) The learning dynamics are fundamentally different (not just LSTM vs Transformer).**
> > > > In RUSP, the agent is trained online with PPO self-play, while the reward structure is randomized at the beginning of every episode. By learning across many such modified incentive settings, the resulting policy becomes robust to different payoff conditions (e.g., when partners may help or act adversarially). However, the policy does not perform test-time or cross-episode adaptation.
> > > >
> > > > CooT, in contrast, uses offline supervised pretraining to apply in-context learning to coordination. By training a transformer on best-response trajectories, CooT learns to infer a partner’s behavior pattern directly from interaction history. As a result, it performs **test-time adaptation without gradients as episodes accumulate** (Section 5.4.1) and remains **robust to abrupt partner behavior changes** (Section 5.4.2).
> > > >
> > > > Crucially, these capabilities arise from sequence modeling and context inference, not from architecture choice alone. Simply swapping RUSP’s LSTM for a Transformer would not yield in-context adaptation, because RUSP learns via reward-shaping–driven PPO updates, whereas CooT learns from trajectory context under the fixed cooperative reward. The two methods depend on fundamentally different learning signals.
> > > >
> > > > **In summary**, RUSP’s online PPO self-play training and focus on social dilemma reciprocity do not reflect the ad-hoc teamwork challenges we study, and its training pipeline violates the requirements of Overcooked or GRF. We therefore treat RUSP as a slightly related but non-comparable baseline.
> > > >
> > > > >"8 episodes for human-AI interaction" - do the authors mean that the same agent played a human for 8 episodes, or that it saw the human play 7 episodes, potentially with other agents, and then used those trajectories as context when collaborating on the 8th?
> > > >
> > > > Sorry for the confusion; each agent played with the same human partner for 8 consecutive episodes in our human study.
> > > >
> > > > For the CooT agent, after each of the first 7 episodes was completed, the full trajectory from that episode was appended to CooT's context window.

---

> > > > > ### Author Response · Authors · 2025-12-03
> > > > > **Further Response to Reviewer Krp4 [2/2]**
> > > > >
> > > > > > How did the tool-assisted pass for the qualitative feedback breakdown work?
> > > > >
> > > > > We prompted Gemini 2.5 Flash to determine if a comment had explicit adaptation mentions or if its tone was negative, and to generate a criterion for both. More specifically, the prompts are as follows:
> > > > > 1. Adaptation: “Look at the comments for the algorithms (CooT, BC, MEP, HSP), list out the IDs of comments that have explicit mentioning of adaptation, and give a rate of negative comments for each algorithm. Please first think about your criteria, and then go over all of them one by one.”
> > > > > 2. Negative: “Look at the comments for the algorithms (CooT, BC, MEP, HSP), list out the IDs of negative comments, and give a rate of negative comments for each algorithm. If the comment mentions improvements, ignore it. Please first think about your criteria, and then go over all of them one by one.”
> > > > > The included comments were later checked manually by a human to avoid false positives, and then the excluded ones to avoid false negatives.
> > > > >
> > > > > Additionally, there is a typo in the original rebuttal we posted; CooT’s number of negative comments should be 13, not 12. The reported percentage (36.1%) is correct.
> > > > >
> > > > > > Experiments on additional environments.
> > > > >
> > > > > As suggested by the reviewer, to strengthen the evaluation, a follow-up experiment on Google Research Football (GRF) was conducted. We now include experiment details and results comparing CooT with the strongest Overcooked baselines, HSP and MEP, in Appendix E. Across all seeds and evaluation partners, CooT achieves a higher goal rate (the number of successful goals within each episode), with a substantial gap over both baselines. This aligns with the trend we observed in Overcooked: **as coordination demands increase, CooT’s advantage becomes more pronounced.**
> > > > >
> > > > > | Method | Goal Rate |
> > > > > | :- | :- |
> > > > > | MEP | $0.75 \pm 0.12$ |
> > > > > | HSP | $0.90 \pm 0.10$ |
> > > > > | **CooT (Ours)** | $\mathbf{1.35 \pm 0.26}$ |
> > > > >
> > > > > These findings directly address the reviewer’s question about general applicability. GRF is a far more demanding cooperative setting than Overcooked, requiring 3-agent teamwork and featuring a more complex, continuous state space (unlike Overcooked's discrete space) and a larger, diverse action space. Even under these challenges, CooT continues to perform strongly, indicating that its coordination ability transfers robustly to more complex multi-agent environments.

---

### Official Review · Reviewer_MpAU · 2025-10-31

**Soundness:** 2
**Presentation:** 2
**Contribution:** 1
**Rating:** 2
**Confidence:** 5

**Summary:**

The paper proposes Coordination Transformers (CooT), a framework for in-context coordination in multi-agent systems. CooT leverages interaction histories to infer and adapt to new partner behaviors on the fly. The model is trained on trajectories collected from diverse agents with different preferences, enabling it to learn implicit coordination strategies without explicit supervision or gradient updates at test time. Experiments in the Overcooked benchmark demonstrate that CooT achieves superior performance compared to population-based methods, gradient-based fine-tuning, and Meta-RL-style contextual adaptation, particularly in zero-shot coordination with unseen partners. The method also shows robustness and strong human compatibility.

**Strengths:**

1. The paper is generally well written and easy to follow. The motivation and the problem formulation are clearly stated. The figures are clear, visually consistent, and effectively convey both the architecture and the experimental results.

2. The paper provides an extensive and carefully designed ablation study. The authors include experiments on non-stationary or changing partners, adaptation over multiple interaction episodes. The ablation studies are detailed, demonstrating CooT's capability in partner adaptation.

3. The Human–AI collaboration study assesses how well CooT collaborates with human partners compared to baseline methods. This shows the proposed framework is not only effective in simulated agent settings but also aligns well with human coordination patterns.

**Weaknesses:**

1. **Insufficient literature review**:
The paper overlooks several highly relevant works in adaptive coordination and partner modeling, such as PACE [1], GSCU [2], and LIAM [3]. These works similarly address how to infer a partner’s latent policy or behavioral intent from interaction history and adapt one’s own policy accordingly. In particular, PACE directly studies peer adaptation with context-aware exploration, which is conceptually very close to the proposed “in-context coordination” problem formulation. The omission of these works weakens the paper’s positioning in the broader literature.

2. **Limited contribution.** The core method of CooT, using a transformer-based policy to autoregressively predict next actions from recent interaction histories, follows a standard paradigm in decision-transformer and in-context RL literature. While the implementation appears solid, the conceptual innovation is limited. Moreover, the problem formulation and methodological scope of CooT appear to be a subset of PACE [1]. PACE not only addresses partner adaptation but also generalizes to opponent and mixed-motive adaptation, and introduces the notion of context acquisition via online exploration when informative histories are unavailable. None of these extensions are discussed or contrasted in this paper. Given that PACE explicitly frames and solves a broader problem under the same high-level theme of context-based coordination, the novelty of CooT remains unclear.

3. **Narrow experimental scope**:
The evaluation is restricted to two-player coordination tasks in Overcooked, which does not fully demonstrate diversity. The method’s scalability to more complex multi-agent settings, including partially observable or mixed-motive environments, remains untested. For comparison, PACE evaluates on PO-Overcooked, which explicitly requires context acquisition through exploration, and also considers multi-agent competition and mixed-motive games involving more than two players. Without experiments in such diverse or challenging settings, it is difficult to judge how well CooT generalizes beyond cooperative, fully observable domains.

4. **Missing comparison with LLM-based baselines**:
The paper does not include comparisons to LLM-based partner coordination methods, such as ProAgent [4], which leverage large language models’ reasoning and common-sense capabilities to infer partner intentions and generate cooperative responses. Given their strong zero-shot coordination abilities, they should serve as an important baseline for CooT. The lack of such comparisons limits the paper’s relevance to the broader research trend of foundation models for interactive decision-making.

[1] Fast Peer Adaptation with Context-aware Exploration

[2] Greedy when Sure and Conservative when Uncertain about the Opponents

[3] Agent Modelling under Partial Observability for Deep Reinforcement Learning

[4] ProAgent: Building Proactive Cooperative Agents with Large Language Models

**Questions:**

1. Could the authors clarify the specific conceptual or methodological contribution of CooT beyond existing works such as PACE, which already investigates in-context partner adaptation using interaction histories? Both frameworks appear to share a similar formulation—treating partner adaptation as a contextual adaptation problem where the policy conditions on recent interactions. From the current description, the main distinction seems to be the use of a transformer-based backbone, while PACE also explores transformer encoders in its appendix.

2. Could the authors consider extending the experimental evaluation beyond the two-player cooperative Overcooked domain? Broader tests in partially observable, mixed-motive, or multi-agent (>2 player) environments would better demonstrate the generality and robustness of the proposed framework.

3. Could the authors consider including comparisons with LLM-driven online adaptation methods, such as ProAgent? These approaches represent an increasingly relevant direction in partner modeling and human–AI coordination.

---

> ### Author Response · Authors · 2025-11-22
> **Rebuttal of Authors**
>
> # Response to Reviewer MpAU [1/2]
> We thank the reviewer for raising important questions about  related work coverage, methodological contribution, experimental scope, and missing comparisons to LLM-based coordination methods. Please find the response to your questions below.
>
> > The literature review omits several key works on adaptive coordination and partner modeling, such as PACE, GSCU, and LIAM, that also infer a partner’s latent policy or intent from interaction history.
>
> We recognize the contribution of existing works, such as PACE, which also aim to enable agents to adapt to unseen partners by conditioning on interaction histories rapidly. While we share this high-level goal, our work introduces a fundamentally different conceptual formulation and learning paradigm for solving it.
>
> **PACE**
>
> The most significant difference between PACE and CooT lies in the learning paradigm and the resultant core capabilities of the agent. PACE is a **Meta-RL** method where a context encoder processes the history of interactions to produce a latent vector that is fed into the policy. This fundamental approach places PACE within the established **context-based Meta-RL literature**, similar to baselines like **PEARL** [1], which we have included in **Table 1** as **HSP-meta**. This latent enables the policy to perform fast adaptation to an unknown peer, which is achieved through PPO optimization using the standard task reward. This method's novelty lies in using an intrinsic reward, derived from an auxiliary peer identification task, to actively encourage the exploration required to generate the necessary context for effective adaptation. CooT, on the other hand,  performs **purely offline, gradient-free in-context adaptation**. The policy is trained once via supervised best-response prediction and adapts at inference time solely by conditioning on recent trajectories.
>
> **GSCU**
>
> Instead of solving cooperative problems like CooT, GSCU is explicitly designed to address **competitive** problems and is aimed at minimizing regret by selecting a policy that exploits the opponent when appropriate, while maintaining conservative robustness when uncertain.
>
> **LIAM**
>
> Like both PACE and CooT, LIAM also conditions the agent's policy on the interaction history to facilitate adaptation. The key distinction from CooT is that **LIAM relies on a learned, recurrent, and compressed latent representation** fed into an RL policy, whereas CooT uses a **raw, sequential trajectory input** for a supervised, gradient-free policy. LIAM uses its architecture to infer the partner's internal state, while CooT uses its architecture for in-context action prediction. Moreover, LIAM is trained **online**, where its recurrent encoder serves as a **feature extractor**, and the decision on the action is made by a **separate RL policy**. Yet, CooT is purely **offline**, and the transformer **is the policy itself**.
>
> While methods like PACE also utilize interaction history to achieve adaptation, we believe CooT introduces a key conceptual divergence. We will try our best to generate a direct quantitative comparison between PACE and CooT within the author response timeframe. Also, we have added discussions about those methods in our revised manuscript (see Chapter 2 and Appendix F).
>
> > The conceptual novelty of CooT appears limited.
>
> To our knowledge, CooT's specific realization of **offline**, **RL-free** partner adaptation is not demonstrated by prior work. This makes **CooT** distinct from effective online methods, such as **PACE**, which rely on Reinforcement Learning and direct interaction to learn and execute adaptation. Thus, despite the similarity of using past data as input, the underlying mechanisms are fundamentally different.
>
> > Extending the experimental evaluation beyond the two-player cooperative Overcooked domain.
>
> We agree with the reviewer that evaluating beyond two-agent Overcooked would provide additional insights into the scalability of context-based adaptation. To address this, we have been running experiments in **Google Research Football (GRF)** [2], a setting that differs from Overcooked in several important ways:
> - **Multi-agent team structure** (3 vs. 1 setting), increasing coordination complexity compared to Overcooked tasks.
> - **Larger and more diverse action space**, resulting in a broader range of partner behaviors and coordination styles.
>
> These properties make GRF a compelling complement to Overcooked for testing cooperative generalization.
> We are committed to submitting the completed results and discussion well in advance of the author comment deadline.

---

> > ### Author Response · Authors · 2025-11-22
> > **Rebuttal of Authors**
> >
> > # Response to Reviewer MpAU [2/2]
> > > The paper does not compare against LLM-based partner-coordination methods such as ProAgent.
> >
> > We agree that LLM-based approaches are a promising trend and offer advantages in interpretability. However, they operate in a different setting. ProAgent relies on language-grounded symbolic states and generates high-level skills through multi-round LLM reasoning, whereas our formulation is language-free and requires action-level online adaptation.
> >
> > ProAgent is also not evaluated with real humans. It uses only a **behavior-cloned human proxy**, which is a static, averaged model and does not capture key aspects of genuine human interaction, such as timing variability, hesitations, or online adaptation. These properties are essential for real-time human–AI collaboration, making coordination with a BC proxy substantially simpler than with actual humans.
> >
> > The reliance on a BC proxy also means that ProAgent is never exposed to the timing-sensitive dynamics of interacting with real humans. In such interactions, **low-latency, action-level responsiveness is critical**, as humans continually adjust based on millisecond-scale timing cues. However, ProAgent’s multi-round LLM reasoning and rule-based low-level controller **cannot** react or adapt within ongoing skill execution. In contrast, CooT produces actions at 2.41 ms per step (~415 Hz), enabling the fine-grained responsiveness needed for smooth human–AI cooperation.
> >
> > Given these differences, LLM-based approaches are complementary but not directly comparable to our real-time, language-free action-level adaptation setting. We have included a discussion about LLM-assisted coordination methods in our revised manuscript (see Chapter 2).
> >
> > [1] Rakelly, Kate, et al. "Efficient off-policy meta-reinforcement learning via probabilistic context variables." International conference on machine learning. PMLR, 2019.
> >
> > [2] Kurach, Karol, et al. "Google research football: A novel reinforcement learning environment." Proceedings of the AAAI conference on artificial intelligence. Vol. 34. No. 04. 2020.

---

> > > ### Comment · Reviewer_MpAU · 2025-11-28
> > > **Reply to Authors**
> > >
> > > Thanks for the clarifications and discussions. I look forward to seeing the additional experimental results on GRF if the authors are able to provide them before the deadline. However, my primary concern regarding the contribution and novelty of the work remains unresolved, as there is still a misunderstanding of key related work.
> > >
> > > The authors state that “CooT, on the other hand, performs purely offline, gradient-free in-context adaptation,” distinguishing it from PACE's “context-based Meta-RL.” This interpretation is not accurate. In PACE, reinforcement learning through interaction with different peers occurs only during the offline training phase to learn a contextual adaptation policy. During online interaction, PACE policy proactively explores peers and generates informative contexts, enabling the agent to adjust its behavior in a gradient-free manner based on the updated contexts. Therefore, PACE should also be considered a gradient-free in-context adaptation method, and fundamentally differs from meta-RL algorithms. Additionally, PACE addresses both cooperative and competitive scenarios, while the current paper only focuses on cooperation, making its scope narrower.
> > >
> > > Given these points, I do not see a clear conceptual or methodological novelty over prior work.

---

> ### Author Response · Authors · 2025-12-03
> **Response to Reviewer MpAU**
>
> > **PACE also performs gradient-free in-context adaptation, so the conceptual distinction and novelty claimed for CooT are unclear.**
>
> We thank the reviewer for this helpful clarification. We agree that PACE operates in a gradient-free manner during deployment. To address the concern regarding novelty and performance, we implemented the full PACE algorithm (HSP-PACE), including its peer-classification module and intrinsic exploration rewards, and evaluated it under our Ad-Hoc Teamwork (AHT) protocol.
>
> The empirical results (Table R1) show that while PACE performs reasonably well in simple settings, its adaptation mechanism becomes unstable in coordination-heavy layouts, resulting in a significant drop in performance compared to HSP-meta and CooT. We have added the results of the HSP-PACE baseline and a detailed discussion in our revision paper, Appendix G.
>
> **Table R1: Comparison of Adaptation Methods**
>
> | Layout | Coord. Ring (Reward) | Coord. Ring Multi-recipe (Reward) |
> | :--- | :--- | :--- |
> | **HSP-meta** | 29.84 ± 3.92 | 30.21 ± 1.37 |
> | **HSP-PACE** | 33.94 ± 3.21 | 4.43 ± 9.02 |
> | **CooT (Ours)** | **38.30 ± 3.71** | **45.96 ± 3.99** |
>
> ### **1. Relation to HSP-meta and Failure Analysis**
> The reviewer correctly notes that context-based adaptation appears in prior works. In our framework, HSP-meta captures the core architecture of this class of methods, using an encoder that conditions the policy on recent interaction histories. PACE extends this with two additional mechanisms: a peer classifier and an intrinsic exploration reward.
>
> Our empirical results show that these additions are detrimental in complex coordination settings:
> * **Simple Layouts:** In *Coord. Ring*, HSP-PACE (33.94) slightly outperforms HSP-meta but still lags behind CooT (38.30).
> * **Complex Layouts:** In *Multi-recipe*, HSP-PACE collapses (4.43). Our analysis suggests two reasons:
>     1.  **Disruptive Exploration:** PACE’s intrinsic reward incentivizes the agent to "probe" the partner to reduce uncertainty. In tight coordination tasks, this active information gathering disrupts the workflow (e.g., abandoning a task to test a partner's reaction), sacrificing joint rewards.
>     2.  **Classification Mismatch:** PACE assumes behaviorally distinct partners, as in PO-Overcooked, where each rule-based partner is restricted to a single ingredient, making identities trivially separable. In our setting, partners exhibit partially overlapping behavioral patterns, causing the classifier to overfit spurious cues rather than true partner differences. This results in unstable or incorrect partner identification, which directly disrupts PACE’s adaptation behavior.
>
> ### **2. Conceptual Clarification: RL vs. Sequence Modeling**
> While both methods are gradient-free at test time, CooT introduces a distinct **learning paradigm** that avoids the pitfalls of PACE’s approach.
>
> * **How the adaptation is learned:**
>     * **PACE (RL):** Learns an active exploration policy. By optimizing for designed rewards, the agent learns to take actions that reduce uncertainty about the partner. As our results show, this "information seeking" behavior often incurs a high cost in tasks that require tight coordination.
>     * **CooT (Supervised Sequence Modeling):** CooT follows the in-context RL (ICRL) paradigm: instead of learning to explore, the model minimizes prediction error on best-response trajectories. Prior work[1] suggests that this objective can induce implicit Bayesian posterior inference, enabling the model to adjust its actions toward the corresponding best-response distribution.
>
>
> ### **Summary**
> CooT is not merely a different architecture; it represents a shift from **Reinforcement-Learned Belief Adaptation** (PACE) to **Supervised Sequence Modeling** (CooT). The empirical collapse of PACE in complex settings validates that CooT’s approach, which avoids the "exploration tax" of RL, is more robust and effective for multi-agent coordination.
>
>
>
> [1] Lee, Jonathan, et al. "Supervised pretraining can learn in-context reinforcement learning." Advances in Neural Information Processing Systems 36 (2023): 43057-43083.

---

> > ### Author Response · Authors · 2025-12-04
> > **Response to Reviewer MpAU**
> >
> > As suggested by the reviewer, to strengthen the evaluation, a follow-up experiment on Google Research Football (GRF) was conducted. We now include experiment details and results comparing CooT with the strongest Overcooked baselines, HSP and MEP, in Appendix E. Across all seeds and evaluation partners, CooT achieves a higher goal rate (the number of successful goals within each episode), with a substantial gap over both baselines. This aligns with the trend we observed in Overcooked: **as coordination demands increase, CooT’s advantage becomes more pronounced.**
> >
> > | Method | Goal Rate |
> > | :- | :- |
> > | MEP | $0.75 \pm 0.12$ |
> > | HSP | $0.90 \pm 0.10$ |
> > | **CooT (Ours)** | $\mathbf{1.35 \pm 0.26}$ |
> >
> > These findings directly address the reviewer’s question about the potential challenges of extending the current framework to multi-agent settings. GRF is a far more demanding cooperative setting than Overcooked, requiring 3-agent teamwork and featuring a more complex, continuous state space (unlike Overcooked's discrete space) and a larger, diverse action space. Even under these challenges, CooT continues to perform strongly, indicating that its coordination ability transfers robustly to more complex multi-agent environments.

---

### Official Review · Reviewer_79kx · 2025-11-02

**Soundness:** 2
**Presentation:** 3
**Contribution:** 2
**Rating:** 4
**Confidence:** 4

**Summary:**

This paper introduces a method, Coot, for in-context adaptation with previously unseen teammates. The method first collects trajectories and then queries a BR "expert" to match the policy of the expert under a context. Experiments show that the method outperforms Meta-RL and gradient-based finetuning.

**Strengths:**

(1) The concept of in-context adaptation for multi-agent generalization is interesting. The method's relation to LLM is very interesting.

(2) The way the experiments are set up makes sense to me

**Weaknesses:**

(1) First of all, ZSC (zero-shot coordination, by Treutlein et al., 2021) and AHT (ad-hoc teamwork, by Stone et al., 2010) are two different things. The paper falsely relates itself to ZSC, which refers to training the same algorithm to always converge to the same convention, rather than to AHT, which generalizes to previously unseen teammates.

- Treutlein, J., Dennis, M., Oesterheld, C., & Foerster, J. (2021, July). A new formalism, method and open issues for zero-shot coordination. In International Conference on Machine Learning (pp. 10413-10423). PMLR.

- Hu, H., Lerer, A., Peysakhovich, A., & Foerster, J. (2020, November). “other-play” for zero-shot coordination. In International Conference on Machine Learning (pp. 4399-4410). PMLR.

(2) No repeated experimentation. The mean ± std is only over 50 rollouts of the same training trial, instead of being repeated multiple times.

(3) Miss meta learning baselines like RL^2 and missing discussion Ad Hoc Teamwork works that use such meta-learning methods. Meta-learning methods like RL^2 using SSM or Transformers usually perform well in prior works on Ad Hoc Teamwork in 2019 Overcooked.

**Questions:**

(1)(2)(3) see weakness

(4) The BR is just the BR to the current unknown teammate, right? Then it may or may not be an optimal adaptation to teammates given the context. Can you prove it is optimal in a rigorous way?

---

> ### Author Response · Authors · 2025-11-22
> **Rebuttal of Authors**
>
> # Response to Reviewer 79kx [1/2]
>
> We thank the reviewer for raising questions about problem formulation, experimental protocol, baseline coverage, and best-response optimality. Please find the response to your questions below.
>
> > ZSC and AHT are distinct problem settings.
>
> We thank the reviewer for highlighting this important distinction. We do not position our method as a zero-shot coordination (ZSC) algorithm; it belongs to the ad-hoc teamwork (AHT) / partner-generalization setting.
>
> ### Why our setting is not ZSC
> Under the formal definition used in [1] and [2], ZSC assumes that all agents are **independently trained copies of the same learning algorithm**. Unseen partners must come from *identical* training pipelines and differ only through random factors such as initialization. The core difficulty in ZSC is that such independently trained **algorithmic clones** may break symmetries differently (e.g., choosing different conventions such as “swerve left’’ vs. “swerve right’’), making cross-play unreliable. Our evaluation setup does not satisfy these assumptions. The partners in our experiments are **Stage-1 HSP agents**, which are not in the same training procedures as the evaluated agents; therefore, the formal ZSC assumptions do not hold. This setting aligns with **AHT/ partner-generalization**, where teammates may be heterogeneous.
>
> ### On the use of ZSC-Eval
> We use ZSC-Eval only as a standardized benchmark providing a fixed set of unseen partners. Given the formal ZSC definition above, the commonly used ZSC-Eval protocol is, in fact, an **AHT benchmark**, because its evaluation partner pool (Stage-1 HSP agents) is not generated by identical learning rules as the evaluated agents. The benchmark name reflects a broader community usage of “zero-shot’’ to mean *no test-time finetuning*, which differs from the stricter ZSC definition in [1], [2].
>
> ### Clarifications
> To avoid future ambiguity, we have updated the related work (Chapter 2) and experiment (5.1) sections to:
> - Explicitly separate formal ZSC from AHT.
> - Clarify that our method is an AHT / few-shot coordination approach.
> - Clarify that ZSC-Eval is used only as a benchmark, not as a claim that our setting satisfies formal ZSC.
>
> > No repeated experimentation. The mean ± std is only over 50 rollouts of the same training trial, instead of being repeated multiple times.
>
> We have clarified the caption to avoid confusion.
> For each method, we train **three fully independent trials** with different seeds. Each trial is then evaluated using **50 rollouts**, and the mean of these rollouts yields a **single training trial-level score**.
>
> The reported mean ± std in the paper is therefore computed **across  the three independent trials**.This setup ensures that the reported results represent **robust, reproducible differences across trials**, rather than artifacts of rollout-level randomness. We have corrected the caption and made this evaluation protocol explicit in the revision.
>
>
> > Miss meta learning baselines like RL^2 and discussion on Ad Hoc Teamwork works that use such meta-learning methods.
>
> We recognize the importance of **RL$^2$** as a historical and foundational meta-learning method. Our current set of baselines already includes PEARL, which represents a more recent and highly effective approach to meta-RL, as stated in its paper [3] by emphasizing its sample efficiency, performance, and more. Consequently, PEARL was adopted as our meta-RL baseline ( HSP-meta ) in Table 1. Also, we conducted a thorough literature search for works specifically applying RL$^2$ to 2019 Overcooked. However, we were unable to find any published implementations that utilize it in this specific environment. To help us ensure the completeness and relevance of our comparison against the established body of work, we would be grateful if the reviewer could share prior works that have utilized RL$^2$ or similar methods on the 2019 Overcooked platform for Ad Hoc Teamwork.

---

> > ### Author Response · Authors · 2025-11-22
> > **Rebuttal of Authors**
> >
> > # Response to Reviewer 79kx [2/2]
> >
> > > The best-response computed in the paper is a best-response to the current unknown teammate, not necessarily the globally optimal adaptation for the given context; a rigorous optimality guarantee is not provided.
> >
> > We agree with the reviewer that the “BR” used in our BR-Prox metric is an empirical best response to the specific evaluation partner, rather than a mathematically certified optimal adaptation.
> >
> > Following ZSC-Eval [4], we construct a partner-specific best response for each evaluation partner by **fixing the partner’s policy** π and training a dedicated MAPPO agent against this fixed π until convergence. The converged policy is then treated as that partner’s empirical best response. This is precisely the construction used in ZSC-Eval and subsequent works; we adopt it unchanged to ensure consistency with the established evaluation pipeline rather than proposing our own notion of BR.
> >
> > We do **not** claim that this BR is provably optimal in a rigorous sense. In cooperative environments like Overcooked, obtaining an analytical or certified optimal best response is intractable due to the large state and action spaces, two-agent coupling, and long-horizon coordination structure. To our knowledge, no prior work provides such proofs for Overcooked; instead, ZSC-Eval and related papers also rely on heavily trained RL policies as **strong empirical upper bounds**.
> >
> > Under this standard interpretation, the BR-Prox metric we reported measures how closely an evaluated agent approaches this empirical upper limit for each fixed teammate. While the absolute value (e.g., 0.57) does not match the BR itself, CooT consistently achieves higher BR-Prox than all baselines under the same reference construction. This indicates that, given an identical empirical BR for each “current unknown teammate,” CooT adapts more effectively than competing methods, even though neither we nor prior work can provide a formal optimality proof.
> >
> > [1] Hu, Hengyuan, et al. "“other-play” for zero-shot coordination." International Conference on Machine Learning. PMLR, 2020.
> >
> > [2] Treutlein, Johannes, et al. "A new formalism, method and open issues for zero-shot coordination." International Conference on Machine Learning. PMLR, 2021.
> >
> > [3] Rakelly, Kate, et al. "Efficient off-policy meta-reinforcement learning via probabilistic context variables." International conference on machine learning. PMLR, 2019
> >
> > [4] Wang, Xihuai, et al. "Zsc-eval: An evaluation toolkit and benchmark for multi-agent zero-shot coordination." Advances in Neural Information Processing Systems 37 (2024): 47344-47377.

---

### Official Review · Reviewer_y8QZ · 2025-11-06

**Soundness:** 3
**Presentation:** 3
**Contribution:** 3
**Rating:** 6
**Confidence:** 3

**Summary:**

This paper introduces CooT, a framework that learns context-driven coordination strategies aligned with a partner’s behavioral policy to maximize collaboration effectiveness. CooT is trained on trajectory pairs collected from interactions between behavior-preferring agents with hidden reward biases and their best-response counterparts. Through this data, CooT learns to infer hidden partner preferences from interaction context and to adapt its coordination policy within a few episodes, even when paired with previously unseen partners. The model maintains a fixed-length FIFO context buffer, ensuring that only recent trajectories are retained for adaptation. CooT is trained using a cross-entropy loss to imitate the best-response agent’s actions given the context of partner behaviors, without relying on gradient-based updates during deployment. Evaluation is conducted in the Overcooked environment across five layouts using the ZSC-Eval benchmark, which measures zero-shot coordination with unseen partners. Results show that CooT generally outperforms all baselines, including Behavior Cloning, MEP, HSP, fine-tuned HSP, and Meta-RL variants, particularly in complex coordination tasks. Furthermore, human-AI collaboration studies demonstrate that CooT achieves the highest mean score and top subjective ratings for adaptability and collaboration quality compared to all other methods.

**Strengths:**

1.	The paper tackles an important and underexplored problem of partner-centric coordination rather than task-oriented learning. This focus makes the approach more realistic for real-world multi-agent and human-AI interaction settings, where uncertainty often arises from the partner’s changing behavior rather than from the task itself.
2.	CooT achieves few-shot adaptation without gradient updates, leveraging contextual information from recent interactions to adjust its policy online. This enables the agent to adapt efficiently to unseen partners at test time, making it practical for dynamic coordination scenarios.
3.	Fine-tuning baselines underperform compared to CooT, highlighting that gradient-based adaptation methods are unstable and less effective when partner behavior shifts. This demonstrates the stability advantage of in-context adaptation.
4.	Experimental results show that CooT adapts quickly and robustly to partner behavior changes, typically recovering effective coordination within only a few (≈6) episodes after a new partner or strategy switch.

**Weaknesses:**

1.	While the paper covers most technical aspects, several implementation and evaluation details are not sufficiently explained in either the main text or supplementary material, requiring to refer to prior work. For example, the ZSC-Eval-based evaluation pipeline is only briefly described, with the similarity metric computation and partner selection process largely assumed from the original ZSC-Eval paper.
2.	The paper lacks deeper analysis of the evaluation results in Table 1. For instance, it is unclear why MEP performs slightly better than CooT in the Coordination Ring layout but not in others. More discussion is needed on whether these differences arise from task complexity, coordination demand, or model design characteristics.
3.	Similarly, it does not analyze why HSP and HSP-ft outperform CooT on Coordination Ring and Asymmetric Advantages layouts but not on others. Since HSP uses reward shaping, it should theoretically provide an upper bound on coordination performance. Why is this not observed across all tasks? Is this due to HSP’s reward design?
4.	Providing explicit details of the reward functions used for each Overcooked layout would improve clarity and help interpret the quantitative results.
5.	The paper provides limited information about the specific event-based reward components used to train the behavior-preferring agents. Including this information would improve understanding of the underlying working of CooT and the baseline methods.
6.	The paper lacks qualitative or visual demonstrations showing how CooT’s coordination behavior evolves over episodes as it adapts to partner policies. Without these, it is difficult to interpret how the model’s adaptation works temporally.
7.	It is unclear whether the CooT agent dynamically switches between behaviors learned from different behavior-preferring agents within an episode or maintains one behavior per partner throughout. Qualitative analysis or visualizations could clarify this temporal aspect of adaptation.
8.	The paper evaluates COOT only in the 2 agent Overcooked environment. Demonstrating its performance across different environments, especially those involving longer episodes or varied coordination structures, would show the context-based adaptation framework’s generalization limits and scalability beyond short, discrete cooking tasks. Additionally, it would be valuable to discuss how context-based adaptation might scale beyond dyadic coordination, outlining the potential challenges of extending the current framework to multi-agent settings, even if this is planned as future work.

**Questions:**

Please see weaknesses.

---

> ### Author Response · Authors · 2025-11-22
> **Rebuttal of Authors**
>
> # Response to Reviewer y8QZ [1/2]
>
> We thank the reviewer for raising questions about implementation details, reward design, behavioral analysis, and the broader generalization scope of the proposed context-based adaptation framework. Please find the response to your questions below.
>
> > Several implementation and evaluation details are not sufficiently explained.
>
> We have revsied the paper to include all the missing ZSC-Eval implementation details (e.g., similarity computation and partner selection) in Appendix B.6.
>
> > Lack of deeper analysis of the evaluation results in Table 1.
>
> While the layout-dependent differences in Table 1 are discussed in Section 5.2, we agree that the connection to model design characteristics could be made clearer.
>
> **Simpler layouts (Coordination Ring, Asymm. Adv.)**
>
> These layouts have low task complexity and minimal coordination demand. Because partner behavior varies very little, population-based methods such as MEP and HSP can form representative policy pools that already cover the small space of possible behaviors. Their performance can therefore match or slightly exceed CooT, and CooT’s ability to adapt from accumulated interaction history provides limited additional benefit here.
>
> **Complex layouts (Bothway Coord., Multi-recipe)**
>
> These tasks require reliable role coordination, timing alignment, and multi-goal tracking. In such settings, fixed policy pools and HSP-ft’s gradient-based adaptation often fail to capture the diversity of partner behaviors. In contrast, CooT’s In-context learning mechanism can adapt across episodes by leveraging full trajectory histories, yielding stronger generalization.
> These clarifications have been added to Section 5.2 for completeness.
>
>
> > HSP should theoretically provide an upper bound on coordination performance. Why it does not outerperform CooT? Is it due to HSP’s reward design?
>
>
> HSP’s use of reward shaping does **not** imply an upper bound on coordination performance in our evaluation setting. Importantly, the theoretical upper bound for any method is the best-response policy to each evaluation partner, not the performance achieved by HSP’s two-stage training pipeline.
>
> HSP is a two-stage method: Stage 1 constructs a diverse partner policy pool, and Stage 2 trains an ego agent that has interacted only with this fixed pool. Reward shaping is applied **exclusively in Stage 1** to stabilize MAPPO (multi-agent variant of PPO) training and produce more behaviorally diverse partner policies and it does not provide the ego agent with shaped feedback during coordination.
>
> Consequently, HSP’s final coordination performance depends primarily on:
> - **The representativeness and behavioral diversity** of the Stage-1 partner policy pool.
> - **How effectively Stage-2 cross-training** aligns the ego agent with that fixed set of partners.
>
> Thus, HSP’s reward shaping in Stage 1 improves partner diversity but does not impose an upper bound on coordination quality in Stage-2 evaluation.
>
> Importantly, this limitation also motivates our design of CooT. In practice, constructing a sufficiently representative policy pool is challenging, and RL cross-training over such a pool is computationally expensive and often unstable. By contrast, **in-context coordination** provides a more scalable alternative: instead of relying on a fixed pool or gradient updates, CooT adapts directly from recent interaction trajectories, striking an effective balance between partner diversity and online adaptability.
>
> > Explicit details of the reward functions used for each layout.
>
> During evaluation, all agents use the same environment reward: successful deliveries are rewarded (+20 for three-ingredient deliveries, +10 for two‑ingredient deliveries in multi‑recipe layouts), with no additional rewards.
>
> During training, the behavior-preferring agents are trained with specific event-based rewards to induce diverse behavior styles in the partner population.
>
> We have now added the complete event-based reward table for all layouts in **Table 5 (Appendix B.1)**, including every event type and the exact biased reward designed to train the behavior-preferring agents under different layouts. These reward functions follow the Hidden-Utility Markov Game formulation used in ZSC-Eval and are consistent with the constraints of each layout.
>
> > Further information on specific event-based reward components used to train the behavior-preferring agents.
>
> We defined different sets of event-based rewards according to the characteristics of each layout. For example, the biased agent in Bothway Coordination cannot deliver soup, so we shifted the biased reward from delivering soup to picking up onions or dishes in order to obtain sufficient biased agent policies. In contrast, the biased agent in the multi-recipe Coordination Ring can access not only onions but also tomatoes, so some of the events we selected are related to tomatoes.

---

> ### Author Response · Authors · 2025-11-22
> **Rebuttal of Authors**
>
> # Response to Reviewer y8QZ [2/2]
> > Qualitative or visual demonstrations showing how CooT’s coordination behavior evolves over episodes as it adapts to partner policies.
>
> We agree with the reviewer that a visualization of how coordination evolves over episodes can enhance the interpretability of our analysis regarding temporal adaptation.
>
> However, two factors make trajectory-level visualization in Overcooked difficult to interpret:
>
> **(1) Coordination strategies are hard to quantify.**
> Many meaningful coordination patterns, such as avoidance of blocking or smooth division of roles, do not manifest as simple statistics. We experimented with trajectory summaries such as event counts per episode, but these signals were too coarse to distinguish different coordination styles or reveal gradual improvements.
>
> **(2) The ego agent’s behavior is tightly coupled with the partner.**
> Small variations in the partner’s timing or movement often lead to large, nonlinear differences in the joint trajectory. As a result, it becomes difficult to isolate changes attributable to “CooT’s adaptation’’ from normal fluctuations caused by the partner’s behavior.
> Given these limitations, we instead provide two forms of temporal evidence that are more informative:
> - **Human qualitative feedback**, where participants directly reported how CooT’s collaboration improved across episodes. We have now added a complete list of feedback and a more thorough analysis in Tables 13 and 14, along with Appendix D.3.
> - **Episode-swap experiments** (Table 4) demonstrate that CooT rapidly realigns when the partner policy changes abruptly.
>
> We believe these analyses more clearly capture the temporal dynamics of CooT’s adaptation than raw trajectory visualizations. We hope those clarify the reviewer's concern, and we would be happy to discuss any specific, actionable suggestions from the reviewer on how these experiments can be conducted.
>
> > Whether CooT dynamically switches between behaviors learned from different behavior-preferring agents within an episode or maintains one behavior per partner throughout.
>
> In our framework, CooT performs **episode-level** adaptation: the context buffer is updated only after each complete episode, so the agent does **not** switch between behaviors within a single episode. This design aligns with the structure of our offline dataset, where each context comprises full trajectories from one behavior-preferring partner and its corresponding best response.
>
> Supporting within-episode switching would require reformulating the context representation to update at finer temporal granularity, which is an interesting direction but **outside the scope of the current method**. To substantiate temporal adaptation under our setup, we include **non-stationary partner experiments in Section 5.4.2 (Table 4)**, in which the partner changes abruptly from one episode to the next. CooT consistently adapts within a few episodes, demonstrating that it can realign its behavior when the context changes, even in the face of abrupt shifts.
>
> We have added clarifications in Section 5.4 to make this explicit.
>
> > Extending the current framework to multi-agent settings.
>
> We agree with the reviewer that evaluating beyond two-agent Overcooked would provide additional insights into the scalability of context-based adaptation.
> To address this, we have been running experiments in **Google Research Football (GRF)** [1], a setting that differs from Overcooked in several important ways:
> - **Multi-agent team structure** (3 vs. 1 setting), increasing coordination complexity compared to Overcooked tasks.
> - **Larger and more diverse action space**, resulting in a broader range of partner behaviors and coordination styles.
>
> These properties make GRF a compelling complement to Overcooked for testing cooperative generalization.
> We are committed to submitting the completed results and discussion well in advance of the author comment deadline.
>
> [1] Kurach, Karol, et al. "Google research football: A novel reinforcement learning environment." Proceedings of the AAAI conference on artificial intelligence. Vol. 34. No. 04. 2020.

---

> > ### Author Response · Authors · 2025-12-03
> > **Follow-up Regarding Google Research Football**
> >
> > As suggested by the reviewer, to strengthen the evaluation, a follow-up experiment on Google Research Football (GRF) was conducted. We now include experiment details and results comparing CooT with the strongest Overcooked baselines, HSP and MEP, in Appendix E. Across all seeds and evaluation partners, CooT achieves a higher goal rate (the number of successful goals within each episode), with a substantial gap over both baselines. This aligns with the trend we observed in Overcooked: **as coordination demands increase, CooT’s advantage becomes more pronounced.**
> >
> > | Method | Goal Rate |
> > | :- | :- |
> > | MEP | $0.75 \pm 0.12$ |
> > | HSP | $0.90 \pm 0.10$ |
> > | **CooT (Ours)** | $\mathbf{1.35 \pm 0.26}$ |
> >
> > These findings directly address the reviewer’s question about the potential challenges of extending the current framework to multi-agent settings. GRF is a far more demanding cooperative setting than Overcooked, requiring 3-agent teamwork and featuring a more complex, continuous state space (unlike Overcooked's discrete space) and a larger, diverse action space. Even under these challenges, CooT continues to perform strongly, indicating that its coordination ability transfers robustly to more complex multi-agent environments.

---

### Author Response · Authors · 2025-12-03
**Summary of Author Response**

Dear AC and Reviewers,

We sincerely thank the reviewers for their thoughtful and constructive feedback. Due to the OpenReview issue, we were unable to engage in further discussion. We would be grateful if the AC could consider our detailed responses when evaluating the revision. We believe all major concerns have been addressed. These revisions further clarify our main contribution: an offline pre-trained, gradient-free in-context coordination method that adapts to unseen partners based solely on interaction history.

Below, we summarize reviewer-identified strengths, key clarifications, and revisions to the manuscript.

## Strengths Highlighted by Reviewers

1. Clear problem formulation: Reviewers MpAU, y8QZ, 79kx highlighted the importance of the partner-centric coordination setting and found in-context multi-agent adaptation to be an interesting direction.

2. Well-structured experiments: Reviewers 79kx and MpAU noted the coherent setup and broad ablations (human partners, strategy shifts, multi-episode adaptation).

3. Stable and fast adaptation: Reviewers y8QZ, Krp4 highlighted CooT’s strong performance, recovery within ≈6 episodes, and robustness in partner-swap settings.

4. Strong human–AI coordination: Reviewers MpAUk, Krp4 emphasized CooT’s consistent performance with real humans.

## Key Clarifications Provided in Text Responses

1. Beyond Overcooked? (y8QZ, MpAU, Krp4): We added GRF experiments, a 3-agent, higher-dimensional domain where CooT again outperforms baselines, consistent with the trend that **greater coordination load strengthens CooT’s advantage.**

2. Why HSP is not an upper bound? (y8QZ): HSP is limited by the coverage of its fixed training partner pool. CooT, via in-context learning, adapts to unseen partners by conditioning on interaction histories, even though trained purely on an offline dataset.

3. Why no RL² baseline? (79kx): HSP-meta (PEARL-based) is a more recent and typically stronger meta-RL method than RL², making it the more relevant comparison.

4. Literature scope coverage (MpAU, R4)
    - GSCU, LIAM, RUSP address competitive/social-dilemma settings.
    - CEC depends on procedural tools unavailable in 2019 Overcooked.
    - LLM-assisted methods like ProAgent operate at high-level skill abstractions and are not designed for the real-time, action-level human–AI coordination required in our setting.

5. Novelty vs. PACE (MpAU): Though both methods are gradient-free during deployment, PACE learns adaptation through an RL objective, similar to our HSP-meta baseline, while CooT adapts purely through supervised sequence modeling of best-response trajectories. We implemented a PACE baseline and discovered that PACE’s collaboration became inconsistent in coordination-heavy settings, whereas CooT’s sequence modeling produced stable in-context adaptation with unseen partners.

## Revisions in the Manuscript

1. GRF Benchmark (MpAU, Krp4, y8QZ): added **GRF settings and results (Appendix E)** showing scalability to higher dimension 3-agent coordination.

2. Clarifying Problem Setting (79kx):  revised **Section 2** and **Section 5.1** to distinguish *AHT* from *formal ZSC* and clarify the use of ZSC-Eval.

3. Evaluation Protocol (79kx): documented details in **Appendix B.6** and clarified results average **3 seeds × 50 rollouts**.

4. Related Work Expansion (MpAU, Krp4): added discussions of PACE, GSCU, LIAM, ProAgent in **Section 2** and **Appendix F**.

5. Reward Design (y8QZ): added **event-based reward tables** and hidden-utility descriptions (Appendix B.1).

6. Layout-Dependent Results (y8QZ): expanded Section 5.2 to clarify how baseline performance changes with coordination demands and why CooT performs better in coordination-heavy layouts.

7. Temporal Adaptation Analysis  (y8QZ): clarified in Section 5.4 that CooT adapts at the episode level, and that partner-swap and human-interaction experiments provide the clearest evidence of its temporal adaptation behavior.

8. Human-Study Qualitative Analysis (Krp4): added coded qualitative results and preprocessing details (Appendix D.3).

9. Computational Cost (Krp4): added compute breakdown including dataset cost, ~19 GPU-hours training, and inference latency (Appendix B.4–B.5).

10. PACE Comparison (MpAU): added **HSP-PACE baseline** and discussion in Appendix G.

---

### Meta-Review · Area_Chair_ZoCr · 2026-01-06

**Summary:**

Summary:
This paper proposes CooT (Coordination Transformer), a method that allows agents to coordinate with previously unseen teammates using in-context learning. CooT conditions its actions on recent interaction histories, enabling on-the-fly adaptation without gradient updates or fine-tuning at test time. The model is trained on trajectories from diverse agents with different hidden preferences, allowing it to infer partner behavior implicitly. Experiments on the Overcooked benchmark show that CooT consistently outperforms Meta-RL, population-based methods, and gradient-based fine-tuning, especially in zero-shot coordination with new partners. Human studies further show that CooT is more robust and preferred as a collaborator.

Strengths:
1. The paper addresses an important and underexplored problem of partner-centric coordination.
2. The proposed method achieves few-shot adaptation without gradient updates.
3. The experiments were well designed
4. The paper is well written and easy to follow.
5. The method’s performance is convincing, particularly with real humans.

Weaknesses:
1. Missing several implementation details.
2. The paper lacks analysis of results and qualitative examples.
3. Mischaracterization of the proposed method’s connection to zero-shot coordination. Ad-hoc teamwork seems more relevant.
4. Lack of repeated experiments.
5. Miss meta learning baselines such as RL^2.
6. Insufficient literature review.
7. The core method follows a standard paradigm in decision-transformer and in-context RL literature, and thus, the contribution is limited.
8. The evaluation was limited to two-player coordination tasks in Overcooked. Scalability to more complex multi-agent settings is clear.
9. No comparison against LLM-based partner coordination methods, such as ProAgent.
10. No discussion of computational costs.
11. More like a few-shot setting, but why the claim of a zero-shot setting?

**Reviewer Concerns:**

Most concerns are addressed. However, a few major ones are still outstanding:
1. The unclear setting: zero-shot, ad-hoc, few-shot. The actual implementation seems closer to few-shot. Few-shot does not need to have weight updates, in my opinion. The multi-agent setting is indeed more aligned with ad-hoc teaming as well. So the mention of zero-shot is quite confusing and could be misleading.
2. Evaluation is limited to Overcooked.

Coupled with limited novelty, the author responses have not fully addressed all important concerns.

**Reviewer Scores:**

Reviewer y8QZ: Already positive. Unclear whether they will further raise the score.

Reviewer 79kx: Likely to raise the score as the responses were adequate.

Reviewer MpAU already acknowledged the rebuttal and indicated that the concern about limited novelty and contribution was not addressed convincingly. So they are unlikely to raise the score to above the acceptance threshold.

Reviewer Krp4 has also indicated that they were not convinced by the responses and asked for further clarification. It is not likely that the reviewer will raise the score above the acceptance threshold after considering the follow-up responses.

---

### Decision · Program_Chairs · 2026-01-26

Reject